# Dr. RAW: Towards General High-Level Vision from RAW with Efficient Task Conditioning

**Wenjun Huang**[*]
University of California, Irvine

**Ziteng Cui**[*]
The University of Tokyo

**Yinqiang Zheng**
The University of Tokyo

**Yirui He**
University of California, Irvine

**Tatsuya Harada**
The University of Tokyo, RIKEN AIP

**Mohsen Imani**[†]
University of California, Irvine

## Abstract

We introduce **Dr. RAW**, a unified and tuning-efficient framework for high-level computer vision tasks directly operating on camera RAW data. Unlike previous approaches that optimize image signal processing (ISP) pipelines and fully fine-tune networks for each task, Dr. RAW achieves state-of-the-art performance with minimal parameter updates and frozen backbone weights. At the input stage, we apply lightweight pre-processing steps, including sensor and illumination mapping, along with re-mosaicing, to mitigate data inconsistencies stemming from sensor variations and lighting conditions. At the network level, we introduce task-specific adaptation through two modules: Sensor Prior Prompts (SPP) and task-specific Low-Rank Adaptation (LoRA). SPP injects sensor-aware conditioning into the network via learnable prompts derived from RAW pixel distribution priors, while LoRA enables efficient task-specific tuning by updating only low-rank matrices in key backbone layers. Despite minimal tuning, Dr. RAW delivers superior results across four RAW-based tasks (object detection, semantic segmentation, instance segmentation, and pose estimation) on nine datasets encompassing various light conditions. By harnessing the intrinsic physical cues of RAW alongside parameter-efficient techniques, Dr. RAW advances RAW-based vision systems, achieving both high accuracy and computational economy. The source code is available here.

## 1 Introduction

Photos recorded in RAW format are increasingly adopted in computer vision tasks due to their captured minimally processed sensor responses [40; 25]. Meanwhile, compared with commonly used sRGB (Fig. 2(a)), RAW data maintains higher bit depth and preserves the intrinsic physical information. These advantages, combined with their linear relationship to scene radiance, allow RAW to outperform sRGB images in various downstream visual tasks under real-world complex lighting conditions, including object detection [37; 20; 54; 58], semantic and instance segmentation [15; 12], tracking [49], pose estimation [30] and so on.

To leverage camera RAW images for high-level visual perception, early works often skipped over the image signal processor (ISP) stage, directly using RAW as input for downstream visual tasks [36; 64; 8], which failed to consider the gap between camera RAW images and sRGB pre-trained

---

[*]Equal contribution.
[†]Corresponding author. email: m.imani@uci.edu

39th Conference on Neural Information Processing Systems (NeurIPS 2025).

weights. Since then, various approaches have been proposed to better improve task-specific performance (Fig. 2(b)), including dynamic ISP parameter tuning [59; 57; 50], additional RAW-to-sRGB encoder networks [16; 54], knowledge distillation [33], and visual adapter tuning [15]. However, existing methods mostly focus on optimizing full ISP & model weights for a single downstream task, ignoring efficient tuning and generalizable intrinsic across diverse real-world scenarios and tasks. Consequently, this gap leads to two key challenges: *data inconsistency* and *tuning inefficiency*.

Regarding *data inconsistency*, datasets for different tasks are typically acquired using distinct camera sensors, while camera manufacturers adopt sensors with varying color response characteristics [40; 1; 38]. At the same time, lighting characteristics and environmental conditions during photography can also cause variations in scene illumination [20; 42]. Formally, the captured camera RAW data can be represented as the following equation [5]:

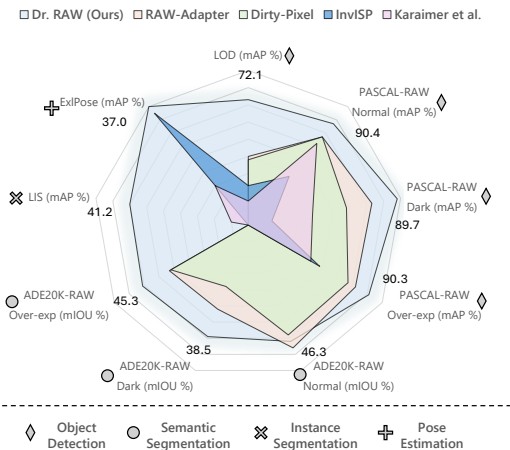

Figure 1: Radar chart demonstrating the superior overall performance of our proposed Dr. RAW.

$$\text{RAW} = \int_{\omega} \rho(x, \lambda) \cdot R(x, \lambda) \cdot L(\lambda) \, d\lambda, \quad (1)$$

where $\omega$ denotes the visible light spectrum (380∼720 nm), $\rho$ the illuminant spectral power distribution, $L$ the sensor-dependent spectral response, and $R$ the scene response. In practice, even for the same scene, the captured RAW would vary due to differences in $\rho$ and $L$. For perception, the inconsistency in data can make the downstream task challenging [47; 34], and current RAW-based perception models may further exacerbate inconsistency due to their task-oriented neural ISPs [15; 50].

Meanwhile, *tuning inefficiency* arises due to current RAW-based perception models normally fully-tuned for a specific task with a single dataset (e.g., object detection [37; 20]). Both the ISP and backbone parameters are optimized to maximize task-specific performance (Fig. 2(b)). When switching tasks or datasets, training both the ISP and downstream network parameters is typically required. Otherwise, the cross-task performance tends to cause catastrophic degradation in existing RAW-based high-level frameworks (Fig. 2(d)). This critical observation underscores the necessity for developing tuning-efficient RAW processing systems that eliminate the requirement for extensive network parameter adjustments.

In this work, we propose **Dr. RAW**, a training-efficient unified solution that addresses the above challenges. Unlike previous methods that require optimizing a large number of ISP modules and training full backbone networks, Dr.RAW applies only two lightweight and task-relevant preprocessing steps, sensor & illumination mapping and re-mosaicing, while omitting other heavy ISP operations. At the network level, we maximize the utilization of sRGB-pretrained knowledge while substantially reducing trainable parameters, achieved by introducing additional ∼2% of the backbone's parameters and freezing the majority of network weights during adaptation. (Fig. 2(d)).

Our contributions could be summarized as follows:

- We propose Dr. RAW, a new framework for RAW-based vision that achieves strong performance across tasks without end-to-end fine-tuning, instead leveraging a frozen RAW-pretrained backbone and lightweight, modular adaptation.

- Pre-processing blocks and task-specific adapters enable Dr. RAW to perform optimally with high flexibility, while effectively mitigating the biases inherent in camera RAW data.

- We demonstrate the effectiveness of Dr. RAW across **4** representative RAW-based high-level vision tasks under a total of **9** diverse conditions (see Fig. 1). Including object detection [37; 20], semantic segmentation [15], instance segmentation [12] and pose estimation [30]. Our method not only outperforms previous SOTA approaches in accuracy but also achieves superior training efficiency.

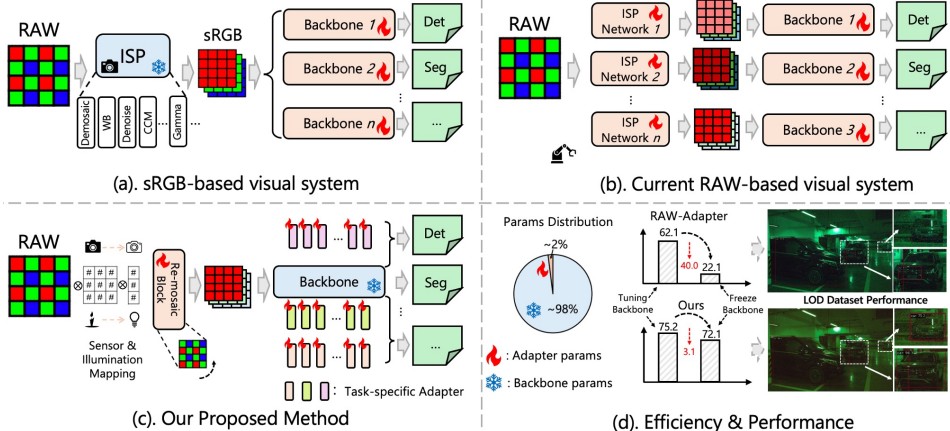

Figure 2: (a). Diagram of sRGB-based visual system. (b). Diagram of the current RAW-based visual system. (c). Our proposed pipeline freezes the backbone parameters and tunes only a few adapter parameters for different tasks. (d). Parameters distribution and tuning & freeze backbone detection performance [20] compare with previous state-of-the-art (SOTA) solution RAW-Adapter [15].

## 2 Related Works

### 2.1 RAW-based Computer Vision Tasks

In recent years, the advantages of camera RAW data have been extensively exploited for various low-level vision tasks such as image denoising [7; 61], super-resolution [56; 62; 29], demoiréing [60; 55], low-light imaging [9; 24], and reflection removal [27]. The rich details in camera RAW images, along with their structured noise distribution, have significantly advanced low-level vision and improved fine-detail reconstruction. Beyond the achievements in image quality improvement, recent advancements have also demonstrated that camera RAW data continues to play an increasingly important role in various high-level machine vision applications.

For RAW-based high-level vision tasks, mainstream approaches either optimize ISP structures and parameters for specific downstream tasks [59; 57; 45; 50; 39] or refine selected intermediate ISP processes [54; 15; 36; 6; 52] (e.g., color correction matrices, look-up tables). For example, ReconfigISP [59] introduces an ISP module pool, then adopts neural architecture search (NAS) to select optimal ISP parameters. AdaptiveISP [50] further enhances this approach by using deep reinforcement learning to adaptively select key ISP parameters. Meanwhile, RAW-Adapter [15] leverages attention mechanisms to optimize parameters and enhance model-level connectivity.

Departing jointly tuning an explicit ISP, research like Dirty-Pixels [16] replaces the ISP with a stack of residual UNets encoder [41]. Chen *et al.* [12] removes the ISP part and additionally adds denoising blocks on the feature map to assist RAW-based instance segmentation. While Li *et al.* [33] distills an inverse ISP pipeline into a new model to improve downstream perception. Despite advancements in RAW-based high-level vision models, existing methods are often fully tuned and overfit for a specific downstream task, lacking the consideration of parameter-efficient task transfer for different tasks.

### 2.2 Parameter-Efficient Fine-Tuning

Parameter-efficient fine-tuning (PEFT) focuses on freezing pre-trained models and either fine-tuning only a subset of network parameters [21; 31; 28] or adding extra parameters for training [10; 23; 22; 26; 46]. Visual prompt tuning (VPT) [23] extends the concept of prompt tuning from natural language processing to computer vision, enabling efficient adaptation of pre-trained models without modifying their core architecture. Instead of altering the parameters in the model, learnable prompts guide the model to adapt to new tasks while preserving its pre-trained knowledge. While [23] introduces task-specific modifications at the input or feature level, low-rank adaptation (LoRA) [22] injects trainable low-rank decomposition matrices into pre-trained weights, optimizing internal weight updates in a low-rank manner. Inspired by these works, we extend such techniques to RAW-based applications.

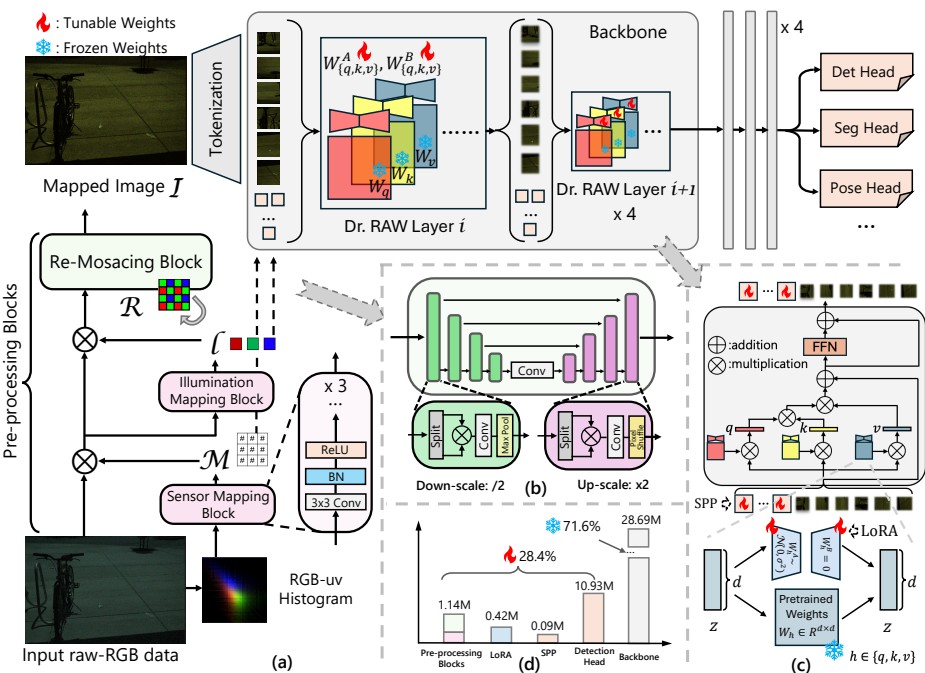

Figure 3: Overview of Dr. RAW. (a). The input RAW is processed by pre-processing blocks and passed to a downstream module with task-specific adapters. (b) Re-mosaicing block in the pre-processing stage. (c) Task-specific adapter design. (d) Parameter distribution across Dr. RAW.

## 3 Method

The overall pipeline of our method is illustrated in Fig. 3 (a). Dr. RAW incorporates pre-processing blocks that map RAW data from various conditions and handle *data inconsistency*. The mapped RAW data is then processed by a versatile backbone augmented with sensor-prior prompts and fine-tuned using LoRA [22] to effectively adapt to downstream tasks. In Sec. 3.1, we detail the design of the pre-processing blocks. In Sec. 3.2, we describe the sensor prior fine-tuning process.

### 3.1 Pre-processing Blocks

Real-world camera imaging systems are subject to continuous variability in both sensor differences and illumination conditions (Eq. 1), which introduces significant changes in pixel distribution, thereby further complicating model optimization across different datasets and tasks [34]. Even data collected by the same camera exhibit considerable variations, as shown by the blue dots in Fig. 4.

To alleviate these variances, we incorporate sensor & illumination mapping blocks followed by a lightweight re-mosaicing block to process input RAW data. Motivated by the white balance design in [2], which first estimates a $3\times3$ matrix $\mathcal{M}$ to eliminate sensor differences and then a $1\times3$ matrix $\mathcal{L}$ for illumination estimation, we adopt a similar two-stage approach. As illustrated in Fig.3 (a) left, we first extract the RGB-uv histogram [18] from the input demosaiced RAW data to get its pixel distribution. The histogram is then fed into a sensor mapping block to estimate the matrix $\mathcal{M}_{3\times3}$, which is then multiplied with RAW data. The transformed image is subsequently passed through an illumination mapping block to estimate the illumination matrix $\mathcal{L}_{1\times3}$. Both sensor mapping and illumination mapping blocks are composed of three simple convolution blocks.

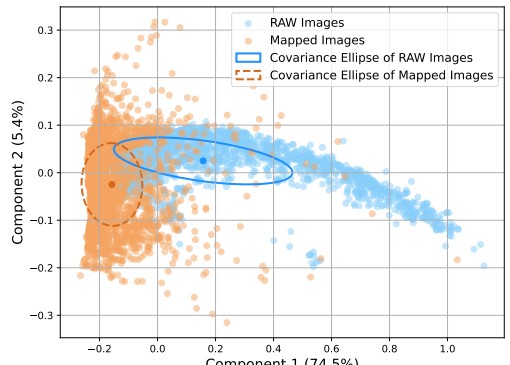

Figure 4: PCA projection of RGB-uv histogram of RAW images [20] and mapped images $\mathcal{I}$.

After that, a re-mosaicing block (Fig.3(b)) $\mathcal{R}$ is further added to alleviate sensor- and scene-dependent artifacts, which is also motivated by findings that reveal unequal contributions of the color channels in camera RAW data [52]. As shown in Fig. 3(b), we design the re-mosaicing block $\mathcal{R}$ as a U-shaped network (details see App. A). We adopt a gating operation, a lightweight nonlinear interaction mechanism that replaces conventional activations. Specifically, the feature map is evenly split along the channel dimension into two halves, and an element-wise product is computed between them. Here, max-pooling is used for down-sampling, while pixel-shuffle [43] is employed for up-sampling at each stage of the U-shaped architecture in a learnable way. The generation of mapped image $\mathcal{I}$ is shown as follows:

$$\mathcal{I} = \mathcal{R}(\text{RAW} \otimes \mathcal{M}_{3\times3} \otimes \mathcal{L}_{1\times3}) \tag{2}$$

We show the PCA projection of histograms in input RAW images and mapped images $\mathcal{I}$ in Fig. 4, it shows that the pre-processing blocks help to reduce the spread of the data distribution during training.

### 3.2 Sensor Prior Efficient Tuning

For the downstream module, current RAW-based visual systems [16; 15; 50] typically require full tuning to avoid performance degradation (see Fig. 2(d)). However, this heavy reliance on updating backbone parameters presents a bottleneck for training efficiency (see Fig. 3(d), backbone accounts for over $70\%$ of the total parameters). To this end, we introduce two simple and effective components that inject task-related knowledge into the backbone in an efficient way.

Taking into account the sensor difference and illumination condition, we propose a sensor prior prompt (SPP) tuning. Specifically, we adopt a set of learnable prompts $\mathcal{P} = \{p_k \in \mathbb{R}^d | k \in \mathbb{N}, 1 \le k \le K\}$ to convey the knowledge gained from the pre-processing block to the backbone of the downstream module. $\mathcal{P}$ is generated by projecting the concatenation of the sensor mapping matrix $\mathcal{M}$ and the illumination mapping matrix $\mathcal{L}$ into a few $d$-dimensional embeddings:

$$\mathcal{P} = FFN([\mathcal{M}_{3\times3}, \mathcal{L}_{1\times3}]) \tag{3}$$

During training, we only fine-tune the $\mathcal{P}$ while keeping the weights in the backbone frozen. Depending on the backbone architecture, we integrate SPP in different ways, which is discussed in App. B.3.

After enabling SPP, the core operation in the transformer, self-attention (SA) (see Eq. 10 in Appendix) in each layer becomes:

$$Attn'(Q', K', V') = \text{softmax}(\frac{Q'K'^T}{\sqrt{d}})V' \tag{4}$$

, where $\mathbf{E}$ is the image patch embeddings, and $Q' = [\mathcal{P}, \mathbf{E}]W_Q, K' = [\mathcal{P}, \mathbf{E}]W_K, V' = [\mathcal{P}, \mathbf{E}]W_V$. Eq. 4 can therefore be decoupled as:

$$Attn'(Q', K', V') = \text{softmax}(\frac{1}{\sqrt{d}} \begin{bmatrix} \mathcal{P}W_Q(\mathcal{P}W_K)^T & \mathcal{P}W_Q(\mathbf{E}W_K)^T \\ \mathbf{E}W_Q(\mathcal{P}W_K)^T & \mathbf{E}W_Q(\mathbf{E}W_K)^T \end{bmatrix}) \begin{bmatrix} \mathcal{P}W_V \\ \mathbf{E}W_V \end{bmatrix} \tag{5}$$

The off-diagonal terms in the attention matrix (i.e., $\mathcal{P}W_Q(\mathbf{E}W_K)^T$ and $\mathbf{E}W_Q(\mathcal{P}W_K)^T$) mean that the SPPs interact with the original image path embeddings in the attention computation, and the top-left term $\mathcal{P}W_Q(\mathcal{P}W_K)^T$ provides sensor-specific influence on the attention. On the other hand, the term $\mathcal{P}W_V$ represents the influence imposed on the original image patch embeddings by SPPs.

In addition to SPP, we further apply LoRA [22] to selected layers of the backbone to enhance task adaptability while preserving efficiency. As illustrated in Fig. 3(c), LoRA injects trainable rank-decomposed matrices $W^A, W^B$ into the attention without modifying the original weights, enabling us to achieve effective fine-tuning with minimal parameter overhead.

$$W'_h = W_h + \Delta W = W_h + W_h^B W_h^A, h \in \{Q, K, V\} \tag{6}$$

To stabilize the tuning, $W^A$ is initialized with a Gaussian distribution, and $W^B$ is initialized with all zeros. By jointly optimizing LoRA and SPP, we retain the benefits of a strong pretrained backbone while introducing task-specific adaptability in a lightweight manner. This hybrid strategy significantly reduces the number of trainable parameters (see Fig. 3(d)) and facilitates fast adaptation to diverse downstream tasks with limited computation. Please find App. B.4 for more details.

# 4 Experiments

## 4.1 Experimental Setup

**Datasets.** We conducted experiments on semantic segmentation, object detection, instance segmentation, and pose estimation, utilizing a combination of various synthetic and real-world RAW image datasets. For object detection, we adopted 2 real-world datasets, PASCAL RAW [37; 15] and LOD [20]. For semantic segmentation, we utilized ADE20K RAW [15]. For instance segmentation, we utilized LIS [12]. As for pose estimation, we used ExLPose [30]. We only used the low-light images of LIS and ExLPose, consistent with other tasks. Refer to App. C for more details.

**Implementation Details.** Dr. RAW is built on the open-source computer vision toolboxes: mmdetection [11], mmsegmentation [13], and mmpose [14]. We conducted comparative experiments with the current SOTA methods. All comparison methods adopt the same data augmentation, mainly including random crop, random flip, multi-scale test, etc. We use mean Intersection over Union (mIoU) to evaluate semantic segmentation, and mean Average Precision (mAP) to evaluate instance segmentation, object detection, and pose estimation performance. The backbone of Dr. RAW is a Swin Transformer tiny (Swin-T) [35]. Since most widely-used backbones are pretrained on RGB images, this introduces a domain gap when applied to RAW images and impacts the performance of downstream tasks. To address this issue, we pretrain the backbone on the large-scale RAW dataset, i.e., AED20K RAW. Once pretrained, the backbone is frozen and transferred to other tasks. Fig. 3(d) presents a statistical breakdown of the parameter count for each component of Dr. RAW.

Refer to App. D for more details. Due to page constraints, we present only the primary results for each task here. Additional results can be found in the corresponding subsections in App. E.

## 4.2 Semantic Segmentation

Tab. 1 provides a comparison of semantic segmentation performance across multiple methods, alongside the parameter efficiency of each model. Traditional ISP-based methods, such as Demosaicing [37] and Karaimer *et al.* [25] show relatively consistent performance under normal and over-exposed conditions, but their performance drops significantly in dark-light conditions. InvISP [53], while competitive in well-lit scenes, deteriorates drastically in the dark, underscoring its sensitivity to illumination variations. Similarly, SID [9] and DNF [24] are designed primarily for low-light conditions and thus only report results for the dark scenario. Among the learning-based alternatives, Dirty-Pixel [16] and RAW-Adapter [15] show improved robustness across lighting conditions. RAW-Adapter, in particular, yields the highest mIoU under normal illumination. However, both methods come with relatively high parameter costs, with RAW-Adapter using 45.16 million parameters, all of which are tunable.

Dr. RAW achieves the best overall performance under challenging illumination. It attains SOTA mIoU in both over-exposed and dark settings, while maintaining competitive results in normal lighting. Notably, Dr. RAW requires fewer tunable parameters—only corresponding to a tunable ratio of 42.9%, which is significantly lower than other competitive approaches. This demonstrates the effectiveness of Dr. RAW's parameter-efficient design, striking a superior balance between segmentation accuracy and model compactness, particularly under diverse lighting conditions.

| Method | No. of params(M)♣ ↓ | mIoU | | |
|---|---|---|---|---|
| | | normal | over-exp | dark |
| Demosacing [37] | | 46.18 | 45.03 | 34.97 |
| Karaimer et al. [25] | | 46.91 | 42.15 | 20.95 |
| InvISP [53] | | 46.08 | 44.06 | 5.02 |
| SID [9] | 44.64 (44.64 / 100%) | - | - | 27.18 |
| DNF [24] | | - | - | 35.86 |
| ROD [54] | | 46.03 | 42.92 | 37.80 |
| Dirty-Pixel [16] | 48.92 (48.92 / 100%) | 46.19 | 44.13 | 36.93 |
| RAW-Adapter [15] | 45.16 (45.16 / 100%) | **46.57** | 44.19 | 37.62 |
| **Dr. RAW** | 47.74 (20.51 / **42.9%**) | 46.29 | **45.28** | **38.46** |

♣ The number of parameters is reported in the format: $x(y/z)$, where $x$ is total, $y$ is tunable, and $z = y/x$.

Table 1: Semantic segmentation results on ADE20K RAW. Best results are **bolded** and second-best are underlined.

Fig. 5 visualizes some examples under various illuminations.

## 4.3 Object Detection

Tab. 2 presents object detection performance on the RASCAL-RAW dataset, while also accounting for model efficiency in terms of tunable parameters. We compare three tuning strategies: *frozen*, where only the detection head is trained with a fixed backbone; *fully-tuned*, where both backbone

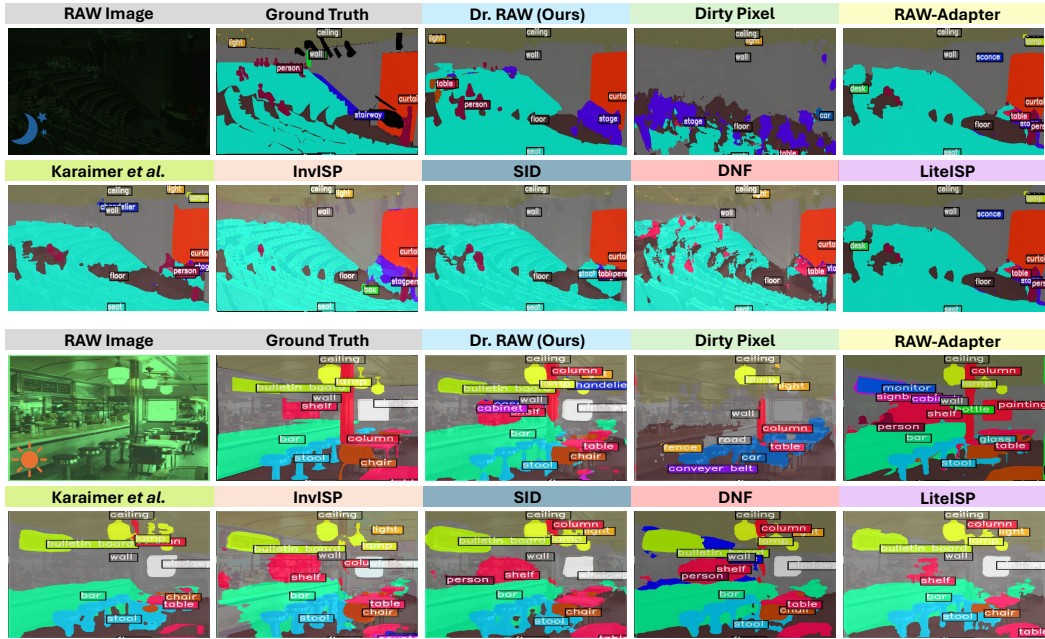

Figure 5: Qualitative comparison of semantic segmentation under different illuminations (low-light 🌙 and over-exposure ☀).

| Method | Setting, No. of params(M)♣ ↓ | mAP | | |
|---|---|---|---|---|
| | | normal | over-exp | dark |
| Default ISP | frozen, 39.62 (10.93 / 28%) | 81.4 | - | - |
| | fully-tuned, 39.62 (39.62 / 100%) | 86.7 | - | - |
| Karaimer et al. [25] | frozen, 39.62 (10.93 / 28%) | 89.3 | 87.6 | 83.1 |
| | fully-tuned, 39.62 (39.62 / 100%) | 90.1 | 89.1 | 87.6 |
| Demosacing [37] | frozen, 39.62 (10.93 / 28%) | 88.1 | 89.4 | 86.5 |
| | fully-tuned, 39.62 (39.62 / 100%) | 90.0 | 90.1 | 87.9 |
| InvISP [53] | frozen, 39.62 (10.93 / 28%) | 88.7 | 89.3 | 72.3 |
| | fully-tuned, 39.62 (39.62 / 100%) | 89.6 | 89.8 | 78.5 |
| Dirty-Pixel [16] | fully-tuned, 39.42 (39.42 / 100%) | 89.7 | 89.0 | 83.6 |
| RAW-Adapter [15] | fully-tuned, 37.11 (37.11 / 100%) | 89.7 | 89.5 | 86.6 |
| **Dr. RAW** | adapter, 38.67 (11.36 / **29%**) | **90.4** | **90.3** | **89.7** |

♣ Refer to Tab. 1 for the format of No. of params.

Table 2: Object detection performance across different methods on RASCAL-RAW (normal / over-exp / dark).

| Method | Setting | mAP |
|---|---|---|
| Default ISP | frozen | 46.8 |
| | fully-tuned | 65.6 |
| Direct (RAW) | frozen | 47.5 |
| | fully-tuned | 67.2 |
| Karaimer et al. [25] | frozen | 40.6 |
| | fully-tuned | 62.5 |
| Dirty-Pixel [16] | fully-tuned | 61.6 |
| RAW-Adapter [15] | fully-tuned | 62.1 |
| **Dr. RAW** | adapter | **72.1** |

Table 3: Object detection performance on LOD.

and head are trained; and *adapter*, our proposed task-conditioned tuning that trains lightweight adapter modules and the detection head while keeping the backbone frozen. Dr. RAW consistently outperforms traditional ISP-based pipelines and recent learning-based methods across all lighting conditions. It maintains a clear advantage over fully-tuned versions of other RAW processing pipelines, particularly exhibiting enhanced robustness in the challenging dark environment where methods like InvISP [53] show marked performance degradation. The baseline Default ISP yields the lowest scores, highlighting the efficacy of specialized RAW domain adaptation. Beyond accuracy, the table provides insights into computational efficiency, specifically focusing on the number of tunable parameters. While possessing a total parameter count (38.67M) comparable to Dirty-Pixel [16] (39.42M) and RAW-Adapter [15] (37.11M), Dr. RAW employs an adapter-based strategy requiring only 11.36M parameters (29% of the total) to be tuned. This contrasts sharply with the others, both of which necessitate tuning 100% of their parameters. Tab. 3 presents the results on LOD. Among all evaluated methods, Dr. RAW achieves the highest mAP. Specifically, Dr. RAW surpasses the strongest fully-tuned baseline by a substantial margin of +4.9% while only adapting a fraction of the model parameters. Qualitative comparison is shown in Fig. 6(a). These results collectively demonstrate that Dr. RAW not only eliminates the need for an ISP and full model tuning but also achieves a new SOTA. The proposed components are both effective and efficient, enabling substantial gains even in the absence of paired supervision or intensive parameter updates.

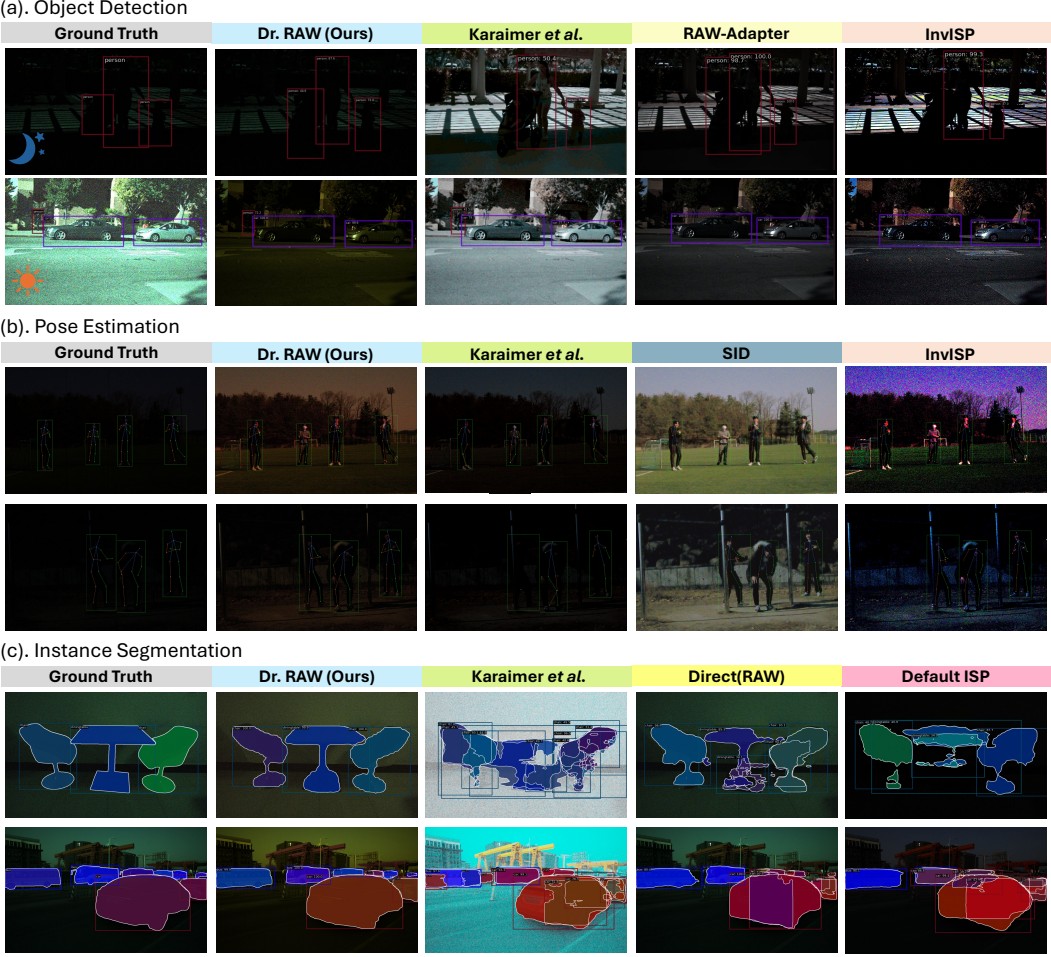

Figure 6: Qualitative comparison of (a). object detection on the PASCAL-RAW [37] dataset under low-light 🌙 and over-exposure ☀ conditions, (b). pose estimation on the ExlPose [30] dataset and (c). instance segmentation on the LIS dataset [12].

## 4.4 Pose Estimation

The quantitative results presented in Tab. 4 report the mAP across several low-light testsets (LL-N, LL-H, LL-E, LL-A), comparing Dr. RAW against relevant prior methods. Operating under the constraint of utilizing only dark RAW images, Dr. RAW consistently establishes a new SOTA benchmark. It achieves superior mAP scores compared to all fully-tuned baselines within this category, including Direct (RAW), Karaimer *et al.* [25], and InvISP [53], across the evaluated low-light testsets. This uniform outperformance underscores the robustness and efficacy of Dr. RAW in extracting salient pose information directly from RAW sensor data, irrespective of the specific low-light challenge. Qualitative comparison is visualized in Fig. 6(b). Additional experimental results against methods leveraging paired RAW-dark and RAW-normal supervision can be found in App. E.

## 4.5 Instance Segmentation

Tab. 5 summarizes the instance segmentation performance of our proposed method, Dr. RAW, against a variety of methods. Within the category trained only on RAW-dark images, Dr. RAW demonstrates compelling performance. The substantial gain in $mAP_{75}$, which demands higher localization accuracy, highlights the quality of the instance masks predicted by our method even under challenging low-light conditions using only dark RAW input. It is noteworthy that Dr. RAW achieves this SOTA performance within the unpaired setting using an efficient adapter-based tuning strategy, rather than requiring full end-to-end fine-tuning like the Direct and Karaimer *et al.* baselines. This underscores the efficacy of our proposed components in Dr. RAW. Fig. 6(c) illustrates representative examples of

instance segmentation results. Additional results against methods leveraging paired RAW-dark and RAW-normal supervision can be found in App. E.

| Method | Setting | mAP by Testset | | | |
|---|---|---|---|---|---|
| | | LL-N | LL-H | LL-E | LL-A |
| Direct (RAW) | frozen | 12.0 | 7.5 | 3.6 | 8.3 |
| | fully-tuned | 36.1 | 26.9 | 18.5 | 29.9 |
| Karaimer et al. [25] | frozen | 9.7 | 6.9 | 3.6 | 7.2 |
| | fully-tuned | 35.4 | 29.2 | 19.1 | 30.1 |
| InvISP [53] | frozen | 6.4 | 3.3 | 2.0 | 4.4 |
| | fully-tuned | 30.5 | 17.7 | 8.2 | 19.9 |
| **Dr. RAW** | adapter | **37.0** | **30.9** | **19.6** | **30.4** |

Table 4: Pose estimation performance across different methods trained solely on RAW-dark images.

| Method | Setting | mAP | $mAP_{50}$ | $mAP_{75}$ | $mAP^{box}$ | $mAP^{box}_{50}$ | $mAP^{box}_{75}$ |
|---|---|---|---|---|---|---|---|
| Default ISP | frozen | 23.3 | 42.3 | 23.0 | 25.8 | 51.4 | 22.9 |
| | fully-tuned | 36.1 | 58.4 | 37.6 | 41.9 | 67.7 | 44.1 |
| Direct (RAW) | frozen | 27.6 | 47.4 | 27.4 | 30.1 | 56.2 | 27.7 |
| | fully-tuned | 40.2 | 61.4 | 41.2 | **44.9** | 70.1 | **48.6** |
| Karaimer et al. | frozen | 18.7 | 34.7 | 18.8 | 20.9 | 43.1 | 17.4 |
| | fully-tuned | 34.6 | 55.1 | 35.5 | 39.7 | 63.9 | 42.4 |
| **Dr. RAW** | adapter | **41.2** | **63.0** | **42.9** | 43.6 | 70.3 | 48.1 |

Table 5: Instance segmentation performance across different methods trained solely on RAW-dark images.

## 4.6 Ablation Study

| Component | | | | Datasets | | |
|---|---|---|---|---|---|---|
| SPP | LoRA | Pre-processing | LOD | PASCAL RAW (normal) | PASCAL RAW (over-exp) | PASCAL RAW (dark) |
| | | ✓ | 43.6 | 81.4 | 86.0 | 59.9 |
| | | ✓ | 57.9 | 88.6 | 89.5 | 81.1 |
| | ✓ | | 69.3 | 88.9 | 89.9 | 89.5 |
| | ✓ | ✓ | 69.8 | 90.3 | 90.1 | 89.4 |
| ✓ | ✓ | ✓ | **72.1** | **90.4** | **90.3** | **89.7** |

Table 6: Component-wise ablation.

| Backbone / Dataset | LOD | PASCAL RAW (normal) | PASCAL RAW (over-exp) | PASCAL RAW (dark) |
|---|---|---|---|---|
| Swin-T (RAW) | **72.1** | **90.4** | **90.3** | **89.7** |
| Swin-T (in1k) | 67.8 | 89.7 | 89.6 | 88.4 |
| ViT (in1k) | 65.9 | 89.5 | 89.4 | 86.6 |

Table 7: Effectiveness of RAW pretraining and generalizability across backbone architectures.

Tab. 6 presents a component-wise ablation study evaluating the impact of the pre-processing block, SPP, and LoRA across the object detection datasets. Without any of these components, performance is significantly lower, particularly under challenging lighting (e.g., 43.6 mAP on LOD and 59.9 on PASCAL RAW (dark)). Introducing the pre-processing block alone yields substantial gains, especially under dark conditions (+21.2), highlighting its effectiveness in stabilizing illumination. Adding LoRA alone also improves results across all datasets, particularly for dark scenes (from 59.9 to 89.5), demonstrating its capacity for efficient adaptation. Combining the pre-processing block and LoRA provides further improvement, especially on LOD (+26.2 over baseline), confirming their complementarity. Finally, integrating VPT with the pre-processing block and LoRA achieves the best results across all datasets, with 72.1 mAP on LOD and over 90 mAP on all PASCAL RAW variants. This indicates that our full model benefits from both robust pre-processing and parameter-efficient tuning mechanisms, achieving consistent gains under diverse lighting conditions.

In addition, we evaluate the performance using three different backbones: Swin-T pretrained on the large-scale RAW dataset (denoted as Swin-T (RAW)), Swin-T pretrained on ImageNet-1k (Swin-T (in1k)), and Vision Transformer pretrained on ImageNet-1k (ViT (in1k)). As shown in Tab. 7, Dr. RAW with Swin-T (RAW) consistently achieves the best performance across all evaluation settings, including LOD and various PASCAL RAW conditions. The substantial performance gap between Swin-T (RAW) and Swin-T (in1k) highlights the importance of pretraining on RAW data, which preserves richer visual information compared to standard RGB inputs. Furthermore, when our techniques are applied to the ViT (in1k) backbone, the model still achieves strong results, demonstrating the generalizability of our proposed components beyond a specific architecture.

| Method | mAP | | | |
|---|---|---|---|---|
| | PASCAL RAW (normal) | PASCAL RAW (over-exp) | PASCAL RAW (dark) | LOD |
| Karaimer et al. (frozen, in1k) | 89.3 | 87.6 | 83.1 | 40.6 |
| Karaimer et al. (fully-tuned, in1k) | 90.0 | 90.1 | 87.9 | 62.5 |
| Direct RAW (frozen, in1k) | 88.1 | 89.4 | 86.5 | 47.5 |
| Direct RAW (fully-tuned, in1k) | 90.0 | 90.1 | 87.9 | 67.2 |
| Dr.RAW (adapter, in1k) | 89.7 | 89.6 | 88.4 | 67.8 |
| Dr.RAW (adapter, RAW) | 90.4 | 90.3 | 89.7 | 72.1 |
| Dr.RAW (fully-tuned, in1k) | 90.6 | 90.4 | 90.2 | 73.8 |

Table 8: Effectiveness of adapter tuning strategy across pretraining datasets.

| Downstream Task Dataset | Backbone Pre-training Dataset | mAP |
|---|---|---|
| LOD | ADE20K (RAW, 20210 images) | 72.1 |
| LOD | PASCAL-RAW (RAW, 4259 images) | 69.3 |
| LOD | in1K (RGB, 1281167 images) | 67.8 |
| LIS | ADE20K (RAW, 20210 images) | 41.2 |
| LIS | PASCAL-RAW (RAW, 4259 images) | 37.8 |
| LIS | in1K (RGB, 1281167 images) | 35.9 |

Table 9: Effect of backbone pre-training dataset on downstream tasks.

Considering the large-scale RAW pretraining is not available for our baselines, we trained Dr. RAW using full fine-tuning on in1K to eliminate the domain mismatch that may bias the results. As shown in Tab. 8, when all methods are fully fine-tuned and initialized with in1K pretrained weights, Dr. RAW significantly outperforms the baseline (Direct RAW). For example, on the LOD dataset, the performance improves from 67.2 to 73.8 (+6.6). In addition, the results also demonstrate that RAW-based pretraining can substantially improve performance. For instance, Dr. RAW improves from 67.8 (in1K-pretrain) to 72.1 (RAW-pretrain), achieving a gain of +4.3. Moreover, compared to fully fine-tuning with In1K pretraining, Dr. RAW reduces the number of trainable parameters by approximately 71%, yet the performance drops by only -1.7 (from 73.8 to 72.1).

The availability of large-scale RAW datasets is a practical consideration. Therefore, we conducted an experiment with the scenario where large-scale synthetic RAW data like ADE20K-RAW is not available, as shown in Tab. 9. We pre-trained a new backbone using only the much smaller PASCAL-RAW dataset (4259 images). We then evaluated this backbone on the LOD and LIS tasks. This study shows that while large-scale pre-training yields the best results, our method still achieves strong performance when pre-trained on a smaller, more accessible RAW dataset, attributing the performance gains come from both the RAW pre-training and our adaptation modules.

To directly evaluate the generalization to unseen sensors, we conducted a zero-shot experiment. We took Dr. RAW trained on the PASCAL RAW dataset (captured with a Nikon DSLR camera) and tested it without any fine-tuning, on RAW images of the same scenes captured with two unseen sensors, iPhone X and Samsung (we adopt [51] to do RAW-to-RAW mapping). The model achieves 88.3 mAP and 88.8 mAP, respectively. Compare with original performance 90.4 on Nikon, the results show only a marginal drop when tested on unseen sensors, demonstrating Dr. RAW's generalization.

### 4.7   Impact of Pre-processing Block

To assess the effectiveness, we analyzed RGB-uv histograms of RAW and mapped images. The RGB-uv histogram captures color distributions in log-chromaticity space [17], where each image is represented by a high-dimensional vector formed by concatenating 2D histograms of the R, G, and B channels over the $(u, v)$ plane [3]. We visualize these histograms using PCA, as shown in Fig. 4. Each point denotes one image's chromaticity distribution, and ellipses illustrate group-wise covariance in the projected space. RAW images exhibit a long, curved spread with a large and anisotropic covariance ellipse, reflecting significant chromatic variability. In contrast, the mapped images form a tight, centered cluster with reduced and more isotropic covariance, demonstrating improved consistency in chromaticity. Quantitatively, the average intra-class distance decreases from 0.380 (RAW) to 0.259 (mapped images), indicating reduced sample variability. The centroids of the two groups are separated by 0.32 in Euclidean distance, confirming a noticeable shift in color representation. The first two principal components explain 74.5% and 5.4% of the variance, capturing the dominant structure of chromatic change. These improvements simplify downstream learning by reducing feature noise and enabling more stable, efficient optimization.

## 5   Conclusion

We present Dr. RAW, a unified and parameter-efficient framework for high-level vision tasks operating directly on RAW images. By combining lightweight sensor-aware pre-processing with modular adapter-based tuning strategies, Dr. RAW achieves SOTA performance across object detection, semantic segmentation, instance segmentation, and pose estimation under diverse lighting conditions. Notably, Dr. RAW minimizes task-specific parameter updates while maintaining robustness and generalizability. Extensive experiments across nine RAW datasets confirm that Dr. RAW effectively bridges the gap between efficient adaptation and high-performance perception in RAW domains.

## Acknowledgements

This work was supported in part by the DARPA Young Faculty Award, the National Science Foundation (NSF) under Grants #2127780, #2319198, #2321840, #2312517, and #2235472, the Semiconductor Research Corporation (SRC), the Office of Naval Research through the Young Investigator Program Award, the U.S. Army Combat Capabilities Development Command (DEVCOM) Army Research Laboratory under Support Agreement No. USMA 21050, and Grants #N00014-21-1-2225 and N00014-22-1-2067. Additionally, support was provided by the Air Force Office of Scientific Research under Award #FA9550-22-1-0253, along with generous gifts from Xilinx and Cisco.

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

# A    Re-mosaicing Block Architecture

The re-mosaicing block (Fig. 3(b)) serves as the fundamental building unit of our model, following a minimalist design philosophy that balances computational efficiency and representational power. Unlike traditional convolutional blocks that heavily rely on complex attention mechanisms or deep non-linearities, the re-mosaicing block adopts a more streamlined yet effective architecture. It leverages simple convolutional operations, channel-wise normalization, and efficient feature modulation strategies to extract and refine features.

A core component in the re-mosaicing block is the use of a simple gating mechanism (SG), a lightweight nonlinear interaction mechanism that replaces conventional activations such as ReLU or GELU. It is inspired by findings that color channels contribute unequally across tasks and lighting conditions [52]. Specifically, the input feature map is evenly split along the channel dimension into two halves, and an element-wise product is computed between them:

$$\text{SG}(x) = x_1 \cdot x_2, \text{where } x = [x1, x2] \tag{7}$$

This operation enables direct and efficient channel-wise interaction without introducing additional parameters or computational overhead. By operating in-place and avoiding expensive non-linear functions, SG significantly reduces memory access costs while still enabling expressive transformations, making it particularly well-suited for real-time and resource-constrained applications.

In the decoding path, the block integrates PixelShuffle [44] for upsampling, which has become a preferred alternative to transposed convolutions due to its artifact-free nature and computational simplicity. PixelShuffle rearranges a tensor of shape $(C \cdot r^2, H, W)$ into a higher-resolution tensor of shape $(C, rH, rW)$, where $r$ is the upscaling factor. This deterministic rearrangement avoids the checkerboard artifacts often introduced by transposed convolutions and preserves fine spatial detail, which is crucial for high-fidelity image enhancement.

The overall structure of the re-mosaicing block is symmetric and modular, consisting of two sequential convolutional segments separated by normalization and nonlinear interactions. This design, free from transformer-style self-attention or heavy MLPs, allows it to be deeply stacked without overfitting or vanishing gradients, making it highly scalable.

# B    Vision Transformer and Swin Transformer

## B.1    Vision Transformer

For a plain vision transformer (ViT) with $N$ layers, an image is divided into $m$ fixed-sized patches $\{I_j \in \mathbb{R}^{3 \times h \times w} | j \in \mathbb{N}, 1 \leq j \leq m\}$, $h, w$ are the height and the width of the image patches. Each patch is then first projected to a $d$-dimensional embedding with positional encoding:

$$e_0^j = \text{Embed}(I_j) \qquad e_0^j \in \mathbb{R}^d, j = 1, 2, \cdots, m \tag{8}$$

We denote the collection of image patch embeddings $\mathbf{E}_i = \{e_i^j \in \mathbb{R}^d | j \in \mathbb{N}, 1 \leq j \leq m\}$, as inputs to the $(i+1)$-th transformer layer $(L_{i+1})$. The ViT is formulated as:

$$\mathbf{E}_i = L_i(\mathbf{E}_{i-1}) \qquad i = 1, 2, \cdots, N \tag{9}$$

Each layer $L_i$ consists of multi-head self attention (MSA) [48] and feed-forward networks (FFN) [4] together with LayerNorm and residual connections [19].

The attention function is computed on the embeddings $\mathbf{E}_i$ packed together into a query matrix $Q = \mathbf{E}_i W_Q$, a key matrix $K = \mathbf{E}_i W_K$, and a value matrix $V = \mathbf{E}_i W_V$, where $W_Q, W_K, W_V \in \mathbb{R}^{d \times d}$. We compute the matrix of outputs as:

$$\text{Attn}(Q, K, V) = \text{softmax}(\frac{QK^T}{\sqrt{d}})V \tag{10}$$

In addition to MSA sub-layers, each of the layers contains a FFN, which is applied to each position separately and identically. It consists of two linear transformations with a ReLU activation in between.

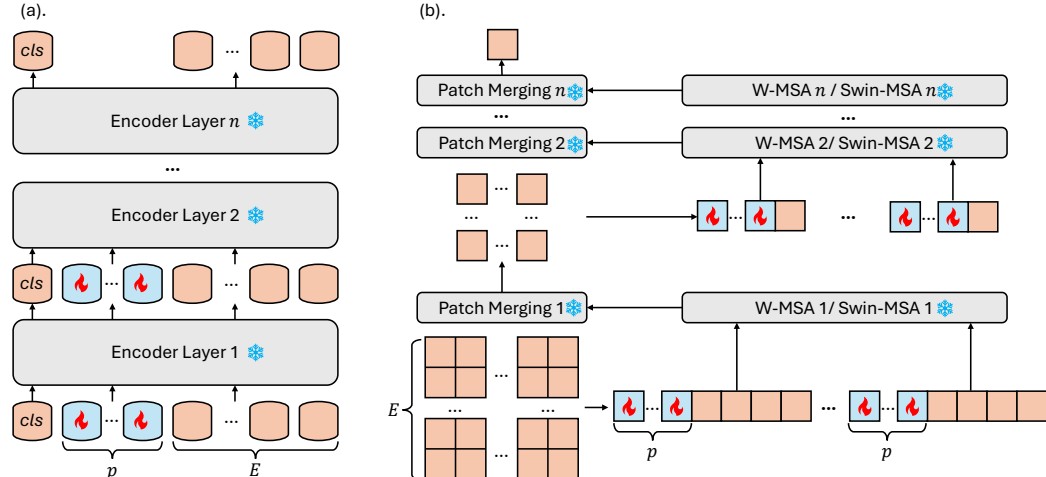

Figure 7: SPP integration. (a). For ViT, we embed the prompts between the $cls$ token and the image embeddings. (b). For Swin Transformer, we insert the visual prompts before Window Multi-Head Self-Attention(W-MSA) and Shifted Window Multi-Head Self-Attention(Swin-MSA), removing it during the patch merging stage.

## B.2 Swin Transformer

Swin transformer [35] is built by replacing the standard MSA module in a transformer block by a module based on shifted windows, with other layers kept the same. Swin transformer computes self-attention within local windows. The windows are arranged to evenly partition the image in a non-overlapping manner. The window-based self-attention module lacks connections across windows, which limits its modeling power. To introduce cross-window connections while maintaining the efficient computation of non-overlapping windows, swin transformer utilizes a shifted window partitioning approach that alternates between two partitioning configurations in consecutive transformer blocks. The first module uses a regular window partitioning strategy that starts from the top-left pixel, and the feature map is evenly partitioned into windows of size $\mathcal{W}$. Then, the next module adopts a windowing configuration that is shifted from that of the preceding layer, by displacing the windows by $(\lfloor \frac{\mathcal{W}}{2} \rfloor, \lfloor \frac{\mathcal{W}}{2} \rfloor)$ pixels from the regularly partitioned windows. The shifted window partitioning approach introduces connections between neighboring non-overlapping windows in the previous layer and is found to be effective in image classification, object detection, and semantic segmentation.

## B.3 Sensor Prior Prompt Integration

We adopt a set of learnable sensor prior prompts (SPP) $\mathcal{P} = \{p_k \in \mathbb{R}^d | k \in \mathbb{N}, 1 \leq k \leq K\}$ to convey the sensor prior knowledge from sensor-independent illumination mapping to the backbone. Here, $K$ denotes the number of SPP adopted in the backbone. During the tuning, only the SPPs are being updated, while the backbone is kept frozen. Each query $p$ is generated by projecting the concatenation of the sensor mapping matrix and the illumination mapping matrix into a few $d$-dimensional embeddings

$$p = FFN([\mathcal{M}_{3\times3}, \mathcal{L}_{3\times3}]) \tag{11}$$

They are inserted in the embeddings after the **Embed** layer (Eq. 8).

For a ViT-based backbone, as shown in Fig. 7(a), we integrate the SPPs between the $cls$ token and image embeddings. The process for the $i^{th}$ layer is formulated as:

$$[cls, , \mathbf{E}_{i+1}] = L_i(cls, p_i, \mathbf{E}_i) \tag{12}$$

, where  denotes removing the token at the position corresponding to $p$ in the output of the $i^{th}$ layer, followed by inserting $p_{i+1}$ before feeding $\mathbf{E}_{i+1}$ into the $(i+1)^{th}$ layer.

For the Swin Transformer-based backbone, we incorporate the SPPs within local windows, excluding them during patch merging, as shown in Fig. 7(b).

## B.4 Low-Rank Adaptation

A typical transformer-based backbone contains many dense layers that perform matrix multiplication (App. B), and the weights in the layers usually have full rank. To efficiently adapt it to a new task, LoRA constrains the update of the weight matrix $W_h \in \mathbb{R}^{d \times d}$ by representing it with a low-rank decomposition, i.e.,

$$W_h' = W_h + \Delta W = W_h + W_h^B W_h^A \tag{13}$$

, where $W_h^A \in \mathbb{R}^{d \times r}$, $W_h^B \in \mathbb{R}^{r \times d}$, and $r \ll d$. During training, $W_h$ is frozen and does not receive gradient updates, while $W_h^A$ and $W_h^B$ contain trainable parameters. Both $W_h$ and $\Delta W$ are multiplied with the same input, and their respective output vectors are summed coordinate-wise. Therefore, consider a matrix multiplication $h = W_h e$ in a well-trained backbone, where $e$ is an embedding, the adapted forward pass yields:

$$h' = W_h e + \Delta W e = W_h e + W_h^B W_h^A e \tag{14}$$

$W_h^A$ is initialized with a Gaussian distribution, and $W_h^B$ is initialized with all zeros, leading to $\Delta W = 0$ at the beginning of the update, and stabilizing LoRA.

In Dr. RAW, we solve the *training inefficiency* by introducing LoRA into the backbone. Specifically, we insert low rank matrices into $W_Q, W_K, W_V$ in Eq. 10, and the weights in FFN. For each task, we train a set of compatible low-rank matrices. Therefore, we can explicitly compute $W_{h_t}' = W_{h_t} + W_{h_t}^B W_{h_t}^A$ for task $t$ during inference. When we need to switch to another downstream task $t'$, we can replace $W_{h_t}^A$ and $W_{h_t}^B$ with $W_{h_{t'}}^A$ and $W_{h_{t'}}^B$, a quick operation with very little memory overhead. Importantly, we do not introduce any additional latency during inference compared to a fine-tuned model by construction.

## C Datasets

We conducted experiments on object detection, semantic segmentation, instance segmentation, and pose estimation, utilizing a combination of various synthetic and real-world RAW image datasets. For object detection, we adopted 2 open-source real-world datasets, PASCAL RAW [37] and LOD [20]. LOD is a real-world dataset consisting of 2230 low-light condition RAW images taken by a Canon EOS 5D Mark IV camera with 8 object classes. We took 1800 images as the training set and the other 430 images as the test set. PASCAL RAW is a normal-light condition dataset with 4259 RAW images, taken by a Nikon D3200 DSLR camera with 3 object classes. Following [15], two synthesized datasets PASCAL RAW (dark) and PASCAL RAW (over-exp) are additionally adopted to verify the generalization capability of Dr. RAW across various lighting conditions. For the semantic segmentation task, we utilized the widely used sRGB dataset ADE20K [63] to generate the RAW dataset with various lighting conditions, namely ADE20K RAW (dark), ADE20K RAW (normal), and ADE20K RAW (over-exp), similar to [15]. The training and test split of ADE20K RAW is the same as ADE20K. For the instance segmentation task, we utilized LIS, containing more than two thousand pairs of low/normal-light images, covering various real-world indoor/outdoor low-light scenes. It includes precise instance-level pixel-wise labels, with a total of 10504 labeled instances across 8 common object classes: bicycle, car, motorcycle, bus, bottle, chair, dining table, and TV. As for pose estimation, we used ExLPose [30], which collected 2556 images of 251 scenes; 2,065 of 201 scenes are used for training, and the remaining 491 of 50 scenes are kept for testing. We only used the low-light images of ExLPose to make the pose estimation consistent with other tasks. Each annotation contains a bounding box and 14 body joints following CrowdPose [32]. An overview of each dataset is presented in Tab. 10.

## D Evaluation Metrics

To evaluate the effectiveness of segmentation models, mean Intersection over Union (mIoU) has become one of the standard metrics due to its robustness and interpretability. Intersection over Union (IoU) quantifies the overlap between predicted and ground truth regions for a given class. It is formally defined as the ratio between the intersection and the union of the predicted and ground truth masks.

$$\text{IoU} = \frac{|\text{Prediction} \cap \text{Ground Truth}|}{|\text{Prediction} \cup \text{Ground Truth}|} \tag{15}$$

| Info.
Dataset | Task | Image Number | Type | Sensor |
|---|---|---|---|---|
| PASCAL RAW [37]
(normal/ dark / over-exp) | Object
Detection | 4259 | real-world & synthesis | Nikon D3200 DSLR |
| LOD [20] | | 2230 | real-world | Canon EOS 5D Mark IV |
| ADE20K RAW [15]
(normal/ dark / over-exp) | Semantic
Segmentation | 27574 | synthesis | - |
| LIS [12] | Instance
Segmentation | 2230 | real-world | Canon EOS 5D Mark IV |
| ExlPose [30] | Pose Estimation | 2556 | real-world | Basler daA1920-160uc
(with Sony IMX392 CMOS) |

Table 10: Overview of the datasets in our experiments.

This formulation penalizes both false positives and false negatives, thus providing a stringent assessment of segmentation quality. Unlike pixel accuracy, which may be overly optimistic in imbalanced datasets, IoU offers a more reliable measure of the model's spatial prediction fidelity. To evaluate performance across multiple semantic categories, IoU is computed for each class individually and then averaged to produce the mean IoU.

$$\text{mIoU} = \frac{1}{C} \sum_{i=1}^{C} \text{IoU}_i \tag{16}$$

This approach ensures that all classes contribute equally to the final score, thereby mitigating the dominance of frequent classes and enabling fairer evaluation in datasets with long-tail distributions.

On the other hand, mean Average Precision (mAP), another widely adopted metric, captures both the precision-recall trade-off and the localization accuracy of model predictions. Unlike accuracy-based metrics, mAP rewards high precision at high recall and penalizes false positives and missed detections, making it a rigorous standard for assessing performance across a range of tasks. For object detection, Average Precision (AP) is computed for each class by integrating the precision-recall curve derived from ranked predictions. A detection is considered correct if it has the correct label and its predicted bounding box achieves an IoU with the ground truth box above a certain threshold. Given a set of predictions sorted by confidence, the precision and recall values are computed at each rank, and the AP for class $c$ is calculated as:

$$\text{AP}_c = \int_0^1 \text{Precision}_c(x) \mathrm{d}x \tag{17}$$

The mean Average Precision (mAP) is then computed as the average over all classes:

$$\text{mAP} = \frac{1}{C} \sum_{i=1}^{C} \text{AP}_i \tag{18}$$

In instance segmentation, mAP is extended by replacing bounding boxes with pixel-level masks. The IoU is thus calculated between predicted and ground truth masks rather than boxes. Accordingly, two metrics are often reported: $\text{mAP}^{box}$ (based on bounding boxes) and $\text{mAP}^{mask}$ (based on instance masks).

For human pose estimation, mAP is computed using the Object Keypoint Similarity (OKS) metric, which measures the similarity between predicted and ground truth keypoints. Unlike IoU, OKS accounts for keypoint visibility and object scale. It is defined as:

$$\text{OKS} = \frac{\sum_i \exp(-\frac{d_i^2}{2s^2 k_i^2}) \delta(v_i = 1)}{\sum_i \delta(v_i = 1)} \tag{19}$$

, where $d_i$ is the Euclidean distance between the predicted and ground truth keypoints, $s$ is the object scale, $k_i$ is a keypoint-specific constant controlling falloff, and $v_i$ is the visibility flag. Similar to mAP in detection, AP is computed at multiple OKS thresholds (e.g., 0.50–0.95), and the final pose mAP is the mean across these thresholds and keypoints.

# E  Detailed Experimental Results

Owing to space constraints, we are unable to include the complete set of experimental results in the main manuscript. Additional results are provided in the appendix. All experiments were conducted on a server equipped with four NVIDIA RTX A6000 GPUs. The software environment includes Python 3.8, PyTorch 1.12, MMDetection 3.3.0, MMSegmentation 1.2.1, and MMPose 1.3.2.

## E.1  Object Detection

Tab. 11 presents per-category AP under varying illumination conditions (i.e., normal, over-exposure (over-exp), and dark), on the PASCAL RAW dataset. Our method, Dr. RAW, consistently achieves the highest AP across all object categories and lighting conditions, demonstrating its robustness to illumination changes. Under normal lighting, Dr. RAWachieves 90.8, 90.3, and 90.1 AP for car, person, and bicycle, respectively, surpassing all competing baselines. Notably, in the challenging dark setting, Dr. RAWoutperforms prior works by large margins, achieving 90.7 (car), 88.6 (person), and 89.7 (bicycle), while the closest runner-up, RAW-Adapter, drops significantly (e.g., only 85.7 for bicycle). While traditional pipelines such as Demosaicing and Karaimer *et al.* perform reasonably under normal and over-exposed conditions, their accuracy degrades in low light. In contrast, InvISP suffers substantial performance drops in the dark (e.g., 74.6 for bicycle), indicating brittleness in extreme scenarios. These results underscore the illumination-invariant capability of Dr. RAWand its effectiveness in learning directly from RAW data without relying on handcrafted ISP operations.

| Method | Normal | | | Over-Exp | | | Dark | | |
|---|---|---|---|---|---|---|---|---|---|
| | Car | Person | Bicycle | Car | Person | Bicycle | Car | Person | Bicycle |
| Dr. RAW | **90.8** | **90.3** | **90.1** | 90.6 | **90.1** | **90.1** | **90.7** | **88.6** | **89.7** |
| Karaimer *et al.* | 90.7 | 89.9 | 89.5 | 90.6 | 87.3 | 89.3 | 89.8 | 85.9 | 87.1 |
| Demosaicing | 90.7 | 89.9 | 89.6 | **90.7** | 89.6 | 90.0 | 89.7 | 86.8 | 87.3 |
| InvISP | 90.4 | 88.6 | 89.8 | 90.6 | 89.5 | 89.3 | 83.5 | 77.5 | 74.6 |
| Dirty-Pixel | 90.6 | 88.3 | 90.0 | 89.9 | 88.7 | 89.2 | 85.5 | 82.8 | 82.6 |
| RAW-Adapter | 90.3 | 88.9 | 89.9 | 90.6 | 88.0 | 89.8 | 89.3 | 84.6 | 85.7 |

Table 11: Per-category performance (mAP) across different illumination conditions and methods on PASCAL RAW.

Tab. 12 reports per-class AP on the LOD dataset. Dr. RAW achieves the best overall balance and outperforms competing methods in 5 out of 8 categories, including chair (81.5), dining table (52.8), and TV monitor (76.0), demonstrating its strong capability in modeling both structural and fine-grained texture details. While RAW-Adapter yields the highest AP on car (91.9), it underperforms on other classes such as bottle (42.5) and TV monitor (42.4), indicating limited generalization. ISP-based pipelines (e.g., Default ISP and Karaimer *et al.*) perform reasonably in structured scenes but degrade on visually complex or texture-sensitive categories like motorbike and dining table. Notably, Direct(RAW) achieves strong results on bottle (72.6) and bus (67.1), but its performance fluctuates due to the lack of the pre-processing block (Sec. 3.1).

In contrast, Dr. RAW not only delivers SOTA results in terms of average AP (72.7), but does so with remarkable parameter efficiency. Our model updates only 29% of the total parameters, significantly reducing storage overhead without sacrificing accuracy. This lightweight fine-tuning strategy proves especially effective in extracting discriminative features directly from RAW inputs while maintaining generalization across varied object types and scenes. These results affirm that our approach successfully bridges the gap between efficiency and performance, setting a new standard for RAW-domain recognition.

## E.2  Pose Estimation

Tab. 13 reports the mAP across several low-light testsets (LL-N, LL-H, LL-E, LL-A), comparing Dr. RAW against relevant prior methods. A distinction is maintained between methods employing paired RAW-dark/RAW-normal supervision ($\diamond$) and those restricted to unpaired training solely on RAW-dark images ($\spadesuit$), the category encompassing Dr. RAW. We highlight the leading performance

| Method | Class | | | | | | | |
|---|---|---|---|---|---|---|---|---|
| | Bicycle | Car | Motorbike | Chair | Dining Table | Bottle | TV Monitor | Bus |
| Dr. RAW | 74.5 | 90.5 | **71.4** | **81.5** | **52.8** | 65.9 | **76.0** | 64.4 |
| Dirty-Pixel | 70.9 | 89.2 | 68.9 | 73.4 | 35.7 | 52.4 | 53.8 | 48.1 |
| RAW-Adapter | 70.4 | **91.9** | 70.6 | 77.9 | 38.0 | 42.5 | 42.4 | 63.4 |
| Default ISP | 76.3 | 90.2 | 63.4 | 79.1 | 41.0 | 63.6 | 51.3 | 59.8 |
| Direct (RAW) | **76.5** | 90.7 | 64.7 | 75.8 | 31.6 | **72.6** | 59.3 | **67.1** |
| Karaimer *et al.* | 72.1 | 89.5 | 61.5 | 73.2 | 28.0 | 63.7 | 52.3 | 59.4 |

Table 12: Per-class performance (AP) across methods on LOD.

| Method | Setting | mAP by Testset | | | |
|---|---|---|---|---|---|
| | | LL-N | LL-H | LL-E | LL-A |
| LLFlow+CPN$^\diamond$ | fully-tuned | 35.2 | 20.1 | 8.3 | 22.1 |
| LIME+CPN$^\diamond$ | fully-tuned | 38.3 | 25.6 | 12.5 | 26.6 |
| DANN$^\diamond$ | fully-tuned | 34.9 | 24.9 | 13.3 | 25.4 |
| AdvEnt$^\diamond$ | fully-tuned | 35.6 | 23.5 | 8.8 | 23.8 |
| Lee et al.$^\diamond$ | fully-tuned | **42.3** | **34.0** | **18.6** | **32.7** |
| Direct (RAW) ♠ | freeze | 12.0 | 7.5 | 3.6 | 8.3 |
| | fully-tuned | 36.1 | 26.9 | 18.5 | 29.9 |
| Karaimer *et al.* ♠ | freeze | 9.7 | 6.9 | 3.6 | 7.2 |
| | fully-tuned | 35.4 | 29.2 | 19.1 | 30.1 |
| InvISP ♠ | freeze | 6.4 | 3.3 | 2.0 | 4.4 |
| | fully-tuned | 30.5 | 17.7 | 8.2 | 19.9 |
| Dr. RAW♠ | adapter | **37.0** | **30.9** | **19.6** | **30.4** |

$\diamond$ Trained with paired RAW-dark/RAW-normal. ♠ Trained solely on RAW-dark images.
Table 13: Pose estimation performance across different methods trained solely on RAW-dark images. Best results are **bolded** and second-best are underlined **within each training category**.

for each metric separately within each training paradigm. Specifically, the best-performing method is indicated in **bold**, and the second-best is underlined. Operating under the constraint of utilizing only dark RAW images, Dr. RAW consistently establishes a new SOTA benchmark. It achieves superior mAP scores compared to all fully-tuned baselines within this category, including Direct (RAW), Karaimer *et al.*, and InvISP, across the evaluated low-light testsets. This uniform outperformance underscores the robustness and efficacy of Dr. RAW in extracting salient pose information directly from RAW sensor data via its adapter-based tuning strategy, irrespective of the specific low-light challenge. Moreover, a comparative analysis against methods leveraging paired RAW-dark and RAW-normal supervision ($\diamond$) reveals the striking competitiveness of Dr. RAW. While the top-performing paired approach (Lee *et al.*) generally exhibits higher mAP, Dr. RAW substantially narrows the performance differential attributable to the supervision type. It surpasses several established paired-data techniques across all conditions. Critically, on the particularly challenging LL-E test set, Dr. RAW's performance marginally exceeds that of Lee *et al.*, suggesting exceptional resilience to extremely low-light scenarios that potentially mitigates the necessity for paired supervision in such demanding contexts.

## E.3 Instance Segmentation

Tab. 14 summarizes the instance segmentation performance of our proposed method, Dr. RAW, against a variety of methods. Our proposed method, Dr. RAW, achieves strong performance while operating in the adapter-based setting, striking a compelling balance between accuracy and parameter efficiency. A critical distinction lies in the training data utilized. Methods marked with $\diamond$ leverage paired RAW-dark and RAW-normal images, providing direct supervision for low-light enhancement or domain translation integrated with the downstream task. In contrast, methods marked with ♠,

| Method | Setting | mAP | mAP$_{50}$ | mAP$_{75}$ | mAP$^{box}$ | mAP$^{box}_{50}$ | mAP$^{box}_{75}$ |
|---|---|---|---|---|---|---|---|
| EnlightenGAN + SGN $^\diamond$ | fully-tuned | 37.1 | 60.2 | 37.4 | 44.5 | 67.0 | 48.6 |
| Zero-DCE + SGN $^\diamond$ | fully-tuned | 36.9 | 60.3 | 37.4 | 44.8 | 67.5 | 49.0 |
| SID $^\diamond$ | fully-tuned | 37.8 | 60.0 | 38.3 | 44.7 | 66.6 | 46.9 |
| REDI $^\diamond$ | fully-tuned | 36.0 | 59.0 | 35.8 | 42.8 | 66.1 | 45.9 |
| Chen et al. $^\diamond$ | fully-tuned | **42.7** | **66.2** | **43.3** | **50.3** | **72.6** | **55.2** |
| Default ISP ♠ | freeze | 23.3 | 42.3 | 23.0 | 25.8 | 51.4 | 22.9 |
| | fully-tuned | 36.1 | 58.4 | 37.6 | 41.9 | 67.7 | 44.1 |
| Direct (RAW) ♠ | freeze | 27.6 | 47.4 | 27.4 | 30.1 | 56.2 | 27.7 |
| | fully-tuned | 40.2 | 61.4 | 41.2 | **44.9** | 70.1 | **48.6** |
| Karaimer *et al.* ♠ | freeze | 18.7 | 34.7 | 18.8 | 20.9 | 43.1 | 17.4 |
| | fully-tuned | 34.6 | 55.1 | 35.5 | 39.7 | 63.9 | 42.4 |
| Dr. RAW ♠ | adapter | **41.2** | **63.0** | **42.9** | 43.6 | **70.3** | 48.1 |

$^\diamond$ The model is trained on the RAW-dark and RAW-normal image pairs.
♠ The model is trained solely on the RAW-dark images.
Table 14: Instance segmentation performance across different methods trained solely on RAW-dark images. Best results are **bolded** and second-best are underlined **within each training category**.

including our Dr. RAW, are trained *solely* on RAW-dark images, representing a more challenging scenario where explicit normal-light guidance is absent during training. For fairness, we highlight the best and second-best results in each training category separately: methods trained with paired RAW-normal supervision ($^\diamond$) and those trained solely on RAW-dark data (♠). Within the category trained only on RAW-dark images, Dr. RAW demonstrates compelling performance. Specifically, Dr. RAW outperforms the strongest baseline in this category, Direct (RAW) fully-tuned, by 1% in mAP, 1.6% in mAP$_{50}$, and 1.7% in mAP$_{75}$. The substantial gain in mAP$_{75}$, which demands higher localization accuracy, highlights the quality of the instance masks predicted by our method even under challenging low-light conditions using only dark RAW input. While the bounding box mAP (mAP$^{box}$) of 43.6 is slightly below the fully-tuned Direct (RAW) method, our mask mAP metrics indicate superior segmentation accuracy. It is noteworthy that Dr. RAW achieves this SOTA performance within the unpaired setting using an efficient adapter-based tuning strategy, rather than requiring full end-to-end fine-tuning like the Direct and Karaimer *et al.* baselines. Compared to methods trained with paired data ($^\diamond$), Dr. RAW is remarkably competitive. While Chen *et al.*, benefiting from paired supervision, achieves the highest overall score (42.7 mAP), Dr. RAW (41.2 mAP) significantly narrows the performance gap. It notably outperforms several paired-data methods like EnlightenGAN+SGN (37.1 mAP), Zero-DCE+SGN (36.9 mAP), SID (37.8 mAP), and REDI (36.0 mAP). This underscores the efficacy of Dr. RAW for robust instance segmentation directly from dark RAW images, achieving results comparable to methods requiring significantly more supervision in the form of paired normal-light images.

Tab. 15 reports per-class performance on LIS for both object detection (AP$^{box}$) and instance segmentation (AP$^{mask}$) across the methods trained solely on RAW-dark images. Dr. RAW consistently outperforms prior approaches across nearly all categories, achieving the best AP$^{box}$ in 6 out of 8 classes and the best AP$^{mask}$ in 7 out of 8 classes. These include significant gains in complex object categories such as motorbike (45.6 box / 50.8 mask), bottle (63.4 / 67.3), and chair (71.7 / 73.3). Compared to Karaimer *et al.* and the Default ISP pipeline, Dr. RAW demonstrates superior adaptability under real-world RAW distributions. While Direct(RAW) benefits from bypassing ISP artifacts, it lacks robustness in categories like TV monitor or car, where subtle color and texture cues are essential. These gains can be attributed to two key components in our architecture. First, the pre-processing block allows the model to normalize global color and exposure shifts across devices and scenes, improving resilience under diverse lighting. In addition, it enables high-fidelity feature extraction early in the pipeline. Second, the parameter-efficient strategies effectively task-specific information without altering the domain-general knowledge in the backbone. Together, these modules bridge the gap between low-level RAW signals and high-level recognition tasks, resulting in strong performance on both detection and segmentation tasks.

| Method | AP$^{box}$ | | | | | | | |
|---|---|---|---|---|---|---|---|---|
| | Bicycle | Chair | Dining Table | Bottle | Motorbike | Car | TV Monitor | Bus |
| Karaimer *et al.* | 34.3 | 66.0 | 30.4 | 59.3 | 37.3 | 27.9 | 19.6 | 42.8 |
| Direct (RAW) | 41.2 | 71.6 | 38.8 | 62.8 | 44.5 | 32.3 | 21.8 | 46.5 |
| Default ISP | 38.0 | 67.3 | 34.9 | 57.8 | 39.7 | 30.4 | 22.1 | 45.2 |
| Dr. RAW | 39.7 | **71.7** | **39.2** | **63.4** | **45.6** | 32.0 | **24.8** | **47.0** |

| Method | AP$^{mask}$ | | | | | | | |
|---|---|---|---|---|---|---|---|---|
| | Bicycle | Chair | Dining Table | Bottle | Motorbike | Car | TV Monitor | Bus |
| Karaimer *et al.* | 21.0 | 66.5 | 24.2 | 62.1 | 38.7 | 12.3 | 4.6 | 47.2 |
| Direct (RAW) | 26.1 | 72.7 | 33.1 | 65.9 | 47.6 | 17.5 | 6.3 | 52.6 |
| Default ISP | 22.7 | 67.3 | 29.5 | 62.2 | 38.7 | 13.4 | 5.1 | 50.0 |
| Dr. RAW | 25.8 | **73.3** | **34.2** | **67.3** | **50.8** | **18.6** | **6.7** | **53.7** |

Table 15: Per-class AP$^{box}$ and AP$^{mask}$ across methods on LIS.

## E.4  Future Direction

For future research directions, we believe it is feasible to train a foundation model based on RAW images that supports multi-task learning without the need for adaptation in each specific task. For example, we could build multiple decoders on a shared backbone to address various RAW-based computer vision tasks. This approach is of crucial importance to real-world systems and downstream tasks such as autonomous driving and wildlife monitoring.

