# OpenReview forum: "Dr. RAW: Towards General High-Level Vision from RAW with Efficient Task Conditioning"
_NeurIPS.cc/2025/Conference — NeurIPS 2025 poster_

### Official Review · Reviewer_mwbm · 2025-07-01

**Clarity:** 4
**Significance:** 3
**Originality:** 3
**Rating:** 4
**Confidence:** 4

**Summary:**

This work proposes Dr. RAW, a framework designed to address data inconsistency and tuning inefficiency in RAW-based visual tasks. The key contributions include a preprocessing mechanism that integrates sensor and illumination mapping modules, followed by a re-mosaicing block, as well as a sensor-prior-efficient tuning strategy that jointly optimizes LoRA and SPP. Experiments on semantic segmentation, object detection, instance segmentation, and pose estimation demonstrate the efficiency of the proposed method.

**Questions:**

My main concerns are outlined in the weaknesses listed above. Please check them.

**Ethical Concerns:**

["NO or VERY MINOR ethics concerns only"]

**Final Justification:**

Considering the pretraining issue, I have decided to maintain my rating.

**Limitations:**

Yes

**Quality:**

3

**Strengths And Weaknesses:**

Strengths:
S1: The motivation is clear and well-justified.
S2: The presentation is well-organized and clearly structured.
S3: The method integrates prompt tuning and LoRA into RAW-based high-level visual tasks, achieving strong performance with parameter-efficient tuning and establishing a new state-of-the-art in this domain.

Weaknesses:
W1: For the object detection task, why are the comparison methods different between the RASCAL-RAW dataset and the LOD dataset? Additionally, further explanation is needed regarding the advantage of Dr. RAW over the fully tuned RAW-Adapter on LOD, as their performance differences are relatively minor on RASCAL-RAW.

W2: As shown in Table 7, the performance benefits significantly from pretraining on the large-scale RAW dataset.
1) It is recommended to provide more details about the pretraining strategy.
2) Do the other comparison methods also leverage RAW-based pretraining? If not, is the comparison fair?

W3: In the definition of learnable prompts in line 160, the variable K should be defined within the main text, rather than only in the appendix.

---

> ### Author Rebuttal · Authors · 2025-07-29
>
> We thank the reviewer for the time and valuable feedback. We found the comments to be very constructive and have noticed several common themes regarding the clarity of our novelty and experimental details.
>
> **Weakness 1**: We appreciate you pointing out the inconsistency in the comparison methods between the two datasets. To address this and provide a more thorough comparison, we have now run experiments for Demosaicing and InvISP on the LOD dataset.
> | Method      | Setting      | mAP  |
> |-------------|--------------|------|
> | Demosacing  | frozen       | 58.5 |
> | Demosacing  | fully-tuned  | 41.1 |
> | InvISP      | frozen       | 56.9 |
> | InvISP      | fully-tuned  | 42.7 |
>
> We will update Table 3 in the final manuscript with these new results to ensure the set of compared methods is consistent across both PASCAL-RAW and LOD.
>
> Your observation about the performance difference is correct and highlights a key strength of our method. The performance gap between Dr. RAW and other methods is more significant on more difficult datasets. PASCAL-RAW is a relatively less challenging dataset where most well-tuned methods can achieve high accuracy. This leads to a "ceiling effect," where the performance differences between top methods are naturally smaller. In contrast, LOD is a significantly more challenging benchmark. This is where the architectural advantages of Dr. RAW become evident, which is reflected in the results: on LOD, Dr. RAW achieves 72.1 mAP, a significant +10.0 mAP improvement over the previous SOTA. This large margin on a challenging, real-world low-light dataset is a strong testament to the superiority of our approach.
>
> **Weakness 2**: Yes, we will add more details about the pre-training. The comparison is fair because all methods were trained on the same RAW datasets; the key difference is our efficient adaptation strategy versus their full fine-tuning approach.
> 1. Pre-training Strategy: We agree that more details on the pre-training strategy are beneficial. Our backbone was pre-trained on the large-scale ADE20K RAW dataset. It was pre-trained specifically to close the domain gap between standard RGB images and RAW sensor data. This was accomplished through an end-to-end training process for the semantic segmentation task on the large-scale ADE20K RAW dataset. After this pre-training phase, the backbone's weights were frozen, and this single, RAW-specialized backbone was then used for all other downstream tasks, such as object detection and pose estimation.
>
> 2. Fairness of Comparison: Thank you for this important question. To clarify:
>     1. Do other methods use RAW-based pre-training? **No.** As stated in our paper, most widely-used backbones are pre-trained on RGB images, which is the standard approach the comparison methods follow before they are adapted to the target RAW datasets.
>     2. Is the comparison fair? **Yes**, the comparison is fair because we're evaluating two different end-to-end methodologies for RAW adaptation, not just the backbones in isolation. While the comparison methods don't use large-scale RAW pre-training, they rely on full fine-tuning to adapt. Prior work's methods start with a standard, RGB-pre-trained backbone and then fully fine-tune the entire network on the target RAW dataset. This full fine-tuning is their mechanism for domain adaptation. Our method proposes a new, more efficient paradigm that includes RAW-specific pre-training to close the domain gap, followed by parameter-efficient fine-tuning on a frozen backbone.
>
> Our paper demonstrates that our proposed paradigm, which includes the RAW pre-training as a key component, is superior to the standard full fine-tuning paradigm.
>
> **Weakness 3**: Thank you for pointing this out. We will revise the manuscript to add a definition for K, as noted in the appendix:
> "Here, K denotes the number of SPP adopted in the backbone."

---

> > ### Comment · Reviewer_mwbm · 2025-08-05
> > **Concerns Regarding RAW Pretraining and Fairness of Comparison**
> >
> > Thank you for your clarification. However, I still have concerns regarding Weakness 2. Specifically, it appears that when your method is pre-trained using RGB images, its performance in Table 7 is only comparable to Direct(RAW) with full fine-tuning. From this perspective, I acknowledge your contribution in achieving competitive results through parameter-efficient tuning.
> >
> > That said, I remain concerned about the feasibility and fairness of the comparisons in Table 2 and Table 3. Your method benefits from pretraining on a large-scale RAW dataset (e.g., AED20K RAW), whereas the other methods rely on RGB-based pretraining. This introduces a domain mismatch that may bias the results in favor of your approach—especially considering that comparable RAW pretraining is not available to other methods.
> >
> > Moreover, although your rebuttal identifies RAW pretraining as a key component of the proposed paradigm, this aspect is not clearly analyzed or emphasized in the main body of the paper. In fact, I could not find any substantial discussion of the pretraining process or its specific impact on performance except the "experiments" section. This lack of explicit analysis makes it difficult to fully assess the role and contribution of RAW pretraining in your overall methodology.
> >
> > Based on the concerns outlined above, I will maintain my current rating at this stage. I look forward to hearing thoughts from other reviewers and the area chairs regarding this pretraining issue.

---

> ### Author Response · Authors · 2025-08-07
>
> Dear reviewer, we thank you for your valuable suggestions and thoughtful review. In Table 2 and Table 3, we compared our adapter tuning method against other fully fine-tuned methods.
>
> To ensure a clear comparison, we have now updated our Dr.RAW results using full fine-tuning on ImageNet-1K. The updated results are shown in the table below:
>
> | Method       | mAP (PASCAL RAW normal) | mAP (PASCAL RAW over-exp) | mAP (PASCAL RAW dark) | mAP (LOD) |
> |-------------|--------|---------|----------|----------|
> | Karaimer *et al.* (frozen, *in1k*)  |  89.3  |   87.6  |  83.1  |  40.6  |
> | Karaimer *et al.* (fully-tuned, *in1k*)  | 90.0 | 90.1  | 87.9   |  62.5  |
> | Direct RAW  (frozen, *in1k*) | 88.1 | 89.4 | 86.5 | 47.5 |
> | Direct RAW  (fully-tuned, *in1k*) | 90.0 | 90.1 | 87.9 | 67.2 |
> | Dr.RAW (adapter, *in1k*) | 89.7 | 89.6 | 88.4 | 67.8 |
> | Dr.RAW (adapter, *RAW*) | 90.4 | 90.3 | 89.7 | 72.1 |
> | Dr.RAW (fully-tuned, *in1k*) | 90.6 | 90.4 | 90.2  | 73.8 |
>
>
> As shown in the updated results, when all methods are fully fine-tuned and initialized with ImageNet-1K (In1K) pretrained weights, our method (Dr.RAW) significantly outperforms the baseline (Direct RAW). For example, on the LOD dataset, the performance improves from 67.2 to 73.8 (+6.6).
>
> In addition, the results also demonstrate that RAW-based pretraining can substantially improve performance. For instance, Dr.RAW improves from 67.8 (In1K-pretrain) to 72.1 (RAW-pretrain), achieving a gain of +4.3. Moreover, compared to fully fine-tuning with In1K pretraining, Dr.RAW reduces the number of trainable parameters by approximately 71%, yet the performance drops by only -1.7 (from 73.8 to 72.1).
>
> We will incorporate these updated results into Table 2 and Table 3 and provide a more detailed RAW-pretrain discussion in the revised version. We sincerely thank you again for your valuable suggestions.

---

### Official Review · Reviewer_cGmR · 2025-07-01

**Clarity:** 1
**Significance:** 2
**Originality:** 3
**Rating:** 4
**Confidence:** 4

**Summary:**

This paper proposes an image processing method to improve the performance of computer vision tasks using RAW images. The approach consists of a preprocessing module with three main blocks and an adapter system. The adapter is a SwinT-based backbone for image processing and takes as input both the preprocessed image and prompts derived from parameters obtained from the preprocessing module. This backbone employs Low-Rank Adaptation (LoRA) techniques tailored to each specific CV task. The method enables learning task-adapted input images with a minimal number of trainable parameters, making the adapter efficient to train for various computer vision applications.

**Questions:**

1. Does the RAW-RGB used as input refer to demosaiced images (where a 1-channel Bayer pattern image has been reconstructed into 3 channels)?

2. Is there potential role overlap between the re-mosaicing block, sensor mapping block, and illumination mapping block?

3. What are the benefits of updating fewer parameters for each task?

4. While the paper evaluates datasets for application scenarios requiring real-time processing, are the processing times actually practical for real-world use?

**Ethical Concerns:**

["NO or VERY MINOR ethics concerns only"]

**Final Justification:**

The rebuttal explanations address my main concerns, and the authors have agreed to implement the requested revisions.
I have therefore updated my recommendation accordingly.

**Limitations:**

Yes

**Paper Formatting Concerns:**

No major formatting issues detected.

**Quality:**

2

**Strengths And Weaknesses:**

Strengths:
- The paper demonstrates high accuracy across multiple computer vision tasks on RAW images.
- The proposal of an integrated framework that directly applies RAW images to multiple CV tasks represents a novel approach.
- It shows the potential for task adaptation with a minimal number of trainable parameters.
- The authors have designed modules to address specific challenges that arise when working with RAW images.
- The approach of combining sensor characteristics with task-specific adaptations is interesting.

Weaknesses:
Clarity:

1. Several important technical explanations are missing:
   - Please add architectural description of the re-mosaicing block
   - Unclear usage of the SG block (only referenced in Fig. 3(b) without explanation)
   - Please add explanation of FFN in Section 3

2. Explanation of figures:
   - Fig. 2(d) lacks clear indication of what the graph represents
   - Reference to Fig. 3(b) alone is inadequate for understanding the structure

3. Please add explanation regarding the training process compromises reproducibility:
   - Please add description of the training process or optimizer
   - Unclear which dataset was used to train W_h and details of LoRA training

Quality:

4. Validation of the effectiveness of pre-processing module:
   - Ablation of the re-mosaicing block and mapping blocks is required to evaluate the design

5. Missing comparisons:
   - The comparison of processing speed or memory efficiency with previous methods is required for application.
   - The comparison of parameter counts in the preprocessing portion with prior work is required for application.

Significance:

6. Practical impact:
   - Task dependency in RAW image processing still requires additional learning
   - Insufficient discussion of processing speed and memory efficiency

Minor Comments:
(1. Introduction) Unclear description
The statement "Otherwise, the cross-task performance tends to cause catastrophic degradation in existing RAW-based high-level frameworks (Fig. 2(d))." is unclear about what this "degradation" refers to.
It seems the authors are likely pointing out that adapters tuned for each task may be inappropriate for other tasks (causing degradation).

(Related works 2.2) Unclear description
The explanation of LoRA is overly simplified. Please clarify what is being low-rank decomposed from the learned weights and how it is being injected into the model.

---

> ### Author Rebuttal · Authors · 2025-07-29
>
> We would like to thank the reviewer for the thoughtful and detailed feedback on our work. We have carefully considered all the comments and provide our point-by-point responses below.
>
> **Weakness 1**: We thank the reviewer for the suggestion and would like to clarify that a detailed breakdown of this component is provided in Appendix A of the submitted manuscript.
> We summarize the key architectural details from the appendix for your convenience:
> 1. Overall Structure: The re-mosaicing block is designed as a symmetric, U-shaped network that balances computational efficiency with representational power.
> 2. Simple Gating (SG): SG is a lightweight and parameter-free nonlinear activation. It splits the input feature map along the channel dimension into two halves and computes their element-wise product without the overhead of conventional activation functions. This design is inspired by findings that color channels contribute unequally across tasks and lighting. It reduces memory costs, making it well-suited for our pipeline.
> 3. PixelShuffle: In the decoding path, the block uses PixelShuffle for upsampling. It avoids checkerboard artifacts and better preserves fine spatial details.
>
> We will ensure that when the re-mosaicing block is introduced in Section 3.1, we more explicitly point the reader to  Appendix A.
>
> We will also add the following description to Section 3 to clarify the role and architecture of FFN:
>
> A transformer layer is composed of two main sub-layers: a Self-Attention (SA) and a Feed-Forward Network (FFN). While the SA is responsible for aggregating information across tokens, the FFN provides a non-linear transformation that is applied to each token's feature independently.
> The FFN architecture is typically composed of two linear layers with a non-linear activation function in between. This design allows the model to process complex features and learn more expressive representations beyond the token-mixing performed by the SA.
>
> **Weakness 2**: We agree that a more explicit explanation, in addition to those in lines 67 - 69, is needed.
> Fig.2(d) is designed to demonstrate the key advantage of Dr. RAW: achieving superior performance while being highly parameter-efficient. It compares Dr.RAW against the previous SOTA, RAW-Adapter, under two different training settings.
> 1. The Y-axis represents object detection performance, measured in mAP on LOD.
> 2. Each bar shows two experimental conditions:
>     1. "Tuning Backbone" represents the standard full fine-tuning, where the entire backbone is trained.
>     2. "Freeze Backbone" represents the backbone is frozen, and only lightweight adapters (~2% of total parameters) are trained (indicated by the pie chart).
> 3. The red number highlights the performance drop when switching from a fully tuned backbone to a frozen one.
>
> RAW-Adapter suffers a performance drop of 40.0 mAP when its backbone is frozen. In contrast, Dr.RAW only has a marginal drop of 3.1 mAP, while still achieving a new SOTA.
>
> The descriptive text for Fig.3(b) is already included in lines 142-150, and we will ensure it's positioned more clearly. It is a visualization of our re-mosaicing block. Please refer to our response to **Weakness 1** for details.
>
> **Weakness 3**: We've included the training details in Section 4.1 and will add the following specifics to the manuscript to ensure clarity.
> 1. Training Process and Optimizer: We used common data augmentation strategies, including random crop, random flip, and multi-scale testing. The optimizer we use is AdamW with a learning rate of 0.0001, betas of (0.9, 0.999), and weight decay of 0.05. We will add a detailed table about the hyperparameters.
> 2. Backbone Pre-training and LoRA Details: Please refer to our response to **Minor 2**.
>
> **Weakness 4**: We kindly refer the reviewer to our response to **Weakness 3** for reviewer **p2T5**.
>
> **Weakness 5**: We've benchmarked our Dr. RAW against the previous SOTA, RAW-Adapter, to evaluate its inference speed and memory usage.
>
> | Method         | Inference Speed | Peak GPU Memory |
> |----------------|------------------|------------------|
> | RAW-Adapter    | 17.5 FPS         | 286 MB           |
> | Dr. RAW (Ours) | 15.9 FPS         | 298 MB           |
>
> Dr. RAW achieves a comparable inference speed and memory usage to the previous SOTA. The modest trade-off in inference speed is accompanied by significantly superior performance and generalization. The primary efficiency gain of Dr. RAW lies in its parameter and storage efficiency. We obtained these SOTA results while fine-tuning only a tiny fraction of the parameters, whereas other methods require tuning 100% of the parameters. This leads to a dramatic reduction in storage costs for deployment. For example, adapting to 5 different tasks would require storing 5 full models for RAW-Adapter, but only 1 shared backbone and 5 tiny adapters for Dr. RAW. This makes our approach far more scalable.
>
> Our pre-processing stage contains 1.14M parameters. The pre-processing module of RAW-Adapter contains 0.58M parameters. While our pre-processing module is larger, it's crucial to understand the different roles these components play in their respective frameworks.
> The module in RAW-Adapter is designed to work in conjunction with a fully fine-tuned backbone. The heavy lifting of adaptation is performed by tuning the entire ~37M parameters. In contrast, ours is designed to create a highly normalized representation that a frozen backbone can effectively use. This leads to enormous overall savings in the total number of tunable parameters required for adaptation.
>
> **Weakness 6**: While Dr. RAW still requires task-specific training to achieve optimal performance, its primary advantage lies in its remarkable efficiency.
> 1. Drastically Reduced Storage: Only lightweight adapters are stored per task.
> 2. Greater Computational Efficiency: The training process requires less GPU memory because fewer gradients need to be computed and stored.
> 3. Faster Adaptation: The model can be adapted to new tasks with significantly less overhead.
>
> **Minor 1**: Existing methods are designed with the assumption that the entire backbone will be fully fine-tuned for a given task. The "catastrophic degradation" refers to the severe performance drop that existing methods suffer when their backbone is frozen, not the misuse of adapters across different tasks. We will revise the sentence for clarity.
>
> **Minor 2**: We thank the reviewer for this comment. A full technical description is available in Appendix B.4, but we agree that a clearer summary in the main text would be beneficial.
> 1. What is being low-rank decomposed?
> LoRA does not decompose the original, pre-trained weights ($W_h,h\\in\\{Q,K,V\\}$​). Instead, it constrains the weight update matrix ($\\Delta W$) that is learned during fine-tuning. It is represented as a low-rank decomposition, specifically as the product of two much smaller matrices:
> $\\Delta W=W_h^B * ​W_h^A​$. Here, $W_h$ remains frozen, and only $W_h^B$ and $​W_h^A$​ are trainable. Consider this example: A pre-trained weight matrix $W_h$​ has dimensions $d\\times d$, fully fine-tuning it would require updating $d^2$ parameters. By choosing a small rank  $r≪d$, we can represent the update $\\Delta W$ using two matrices:  $​W_h^A$​ of size $d\times r$ and $W_h^B$ of size $r\times d$. The total number of trainable parameters for this update is only $(d\\times r)+(r\\times d)=2dr$, which is significantly fewer than the $d^2$ parameters required for full fine-tuning.
> We apply this to the key dense layers of the backbone: specifically, the weight matrices of the self-attention mechanism ($W_Q$​, $W_K$​, $W_V$​).
> 2. How is it being injected into the model?
> It is "injected" into the model as a parallel path during the forward pass. Given an input embedding $e$, the output is calculated by passing $e$ through $W_h$ and $\\Delta W_h$ simultaneously, and then summing their outputs.
> The adapted forward pass is therefore:
> $h^\prime=W_h ​e + \\Delta W e=W_h ​e + W_h^B ​W_h^A ​e$. This allows us to efficiently modify the behavior of a layer without altering the original weights. During inference, the learned update $\\Delta W$ can be merged with $W_h$​ by simple matrix addition, meaning we introduce no additional latency.
>
> **Question 1**: We thank you for the clarifying question. The term "raw-RGB" refers to RAW sensor data that has been converted into a 3-channel RGB format using a simple demosaicing algorithm. This is a standard and necessary pre-processing step adopted by all previous works.
>
> **Question 2**: We kindly refer the reviewer to our response to **Weakness 3** for reviewer **p2T5**.
>
> **Question 3**: It provides three main advantages
> 1. Drastically Reduced Storage Costs: Instead of saving a complete large model for each task, we store the task-specific parameters, which can be less than 1% of the total.
> 2. Improved Training Efficiency: Updating fewer parameters requires less GPU memory to compute and store gradients.
> 3. Better Knowledge Retention: By keeping the backbone frozen, the model avoids "catastrophic forgetting" and retains the powerful, general-purpose knowledge from its initial training.
>
> **Question 4**: Dr. RAW achieves an inference speed of 15.9 FPS. This is highly comparable to RAW-Adapter, which runs at 17.5 FPS. A processing speed of 15.9 FPS is sufficient and practical for a range of applications.

---

> > ### Comment · Reviewer_cGmR · 2025-08-03
> > **One follow-up question**
> >
> > Thank you for the detailed rebuttal and the additional results. Incorporating the explanations you provided into the main paper would make it even stronger.
> >
> > I have one follow-up question: the inference speed you report—does it cover the entire pipeline including the downstream task network, or is it measured only for the pre-processing blocks + the backbone?

---

> > > ### Author Response · Authors · 2025-08-03
> > >
> > > We sincerely thank the reviewer for their positive feedback on our rebuttal. We are glad our explanations were helpful and will incorporate them into the revised manuscript to improve its clarity.
> > >
> > > The reported speed of 15.9 FPS is the end-to-end performance. This measurement covers the entire pipeline, including our pre-processing blocks, the frozen backbone, and the downstream task head.
> > >
> > > We greatly appreciate your constructive engagement throughout this process. We hope that our detailed responses have fully addressed your concerns and have demonstrated the strengths and novelty of our work. We respectfully hope that you might reconsider your evaluation in light of this new evidence.

---

> > > > ### Comment · Reviewer_cGmR · 2025-08-04
> > > >
> > > > Thank you for your comprehensive and detailed rebuttal. It has addressed my main concerns, and I am inclined to raise my recommendation to a Borderline Accept.
> > > >
> > > > My recommendation is contingent on the following revisions being incorporated into the final manuscript:
> > > >
> > > > • integrate rebuttal content: incorporate all the valuable explanations provided in the rebuttal into the paper.
> > > > • define terminology: provide clear definitions for paper-specific terms (e.g., “task head”) upon first use.
> > > > • clarify module roles: for each module, briefly state its purpose and role within the main text before directing readers to the appendix for architectural details.
> > > >
> > > > As this is an application paper, please prioritize explanations that practitioners—including those in industry—can easily follow.

---

> > > > > ### Author Response · Authors · 2025-08-04
> > > > >
> > > > > Thank you for your thoughtful feedback and for considering a Borderline Accept. We appreciate your suggestions and will revise the manuscript accordingly by:
> > > > >
> > > > > Integrating key explanations from the rebuttal; defining all paper-specific terms (e.g., “task head”) at first mention; clarifying each module’s role in the main text before referring to the appendix.
> > > > >
> > > > > We will also refine the writing to ensure better accessibility for readers and practitioners. Thank you again.

---

### Official Review · Reviewer_uAEX · 2025-07-03

**Clarity:** 4
**Significance:** 4
**Originality:** 3
**Rating:** 5
**Confidence:** 4

**Summary:**

This paper proposes Dr. RAW, a modular framework for performing high-level visual tasks (e.g., detection, segmentation, pose estimation) directly on RAW sensor data. Unlike prior approaches that rely on neural ISPs or per-task fine-tuning, Dr. RAW introduces a lightweight preprocessing block (comprising sensor and illumination mapping and re-mosaicing), followed by a frozen vision backbone enhanced with SPP and LoRA-based adapters. These modules together normalize domain shifts across sensors and environments and enable efficient per-task adaptation with minimal parameter updates. In experiments, the paper demonstrated both robustness and efficiency on various tasks/datsets.

**Questions:**

I would appreciate if authors could address some unclear points:

- The paper introduces a re-mosaicing block with simple gating as a key component of the preprocessing pipeline. However, no ablation is provided to justify the necessity of re-mosaicing or gating. Could the authors include comparisons with simpler alternatives, such as direct demosaicing using fixed Bayer patterns, to isolate the contribution of this component?

- One of the main claims is that the preprocessing block unifies sensor and illumination variation. Currently, this is supported mainly by qualitative visualizations (e.g., PCA of chromaticity). Could the authors provide quantitative measures (e.g., feature distribution divergence, domain classification accuracy) to substantiate this claim?

- The paper implies that the preprocessing block is shared across tasks and does not need to be retrained for each new task. However, this block depends on scene illumination and sensor response. Could the authors clarify under what conditions the parameters remain reusable, and whether domain-specific fine-tuning is ever required?

- The sensor prompt is generated from a pair of small matrices representing sensor and illuminant characteristics. This hand-crafted formulation seems restrictive. Have the authors explored learned or higher-dimensional embeddings for prompts? Is there a trade-off in interpretability or generalization?

- While the model is tested on multiple datasets, it is unclear how well it generalizes to sensors or conditions not seen during training. Have the authors tested cross-domain adaptation (e.g., training on some sensors and evaluating on held-out ones)? If not, would they expect the system to remain robust under such scenarios?

**Ethical Concerns:**

["NO or VERY MINOR ethics concerns only"]

**Final Justification:**

I have read the authors' feedback and the discussion between the authors and other reviewers thoroughly, and I have found that most of my concerns have been resolved. Therefore, I recommend the acceptance of this paper.

**Limitations:**

yes

**Quality:**

4

**Strengths And Weaknesses:**

Strengths:
I basically found this paper quite insightful because,
- The paper addresses a practical problem how to use RAW sensor data directly for high-level tasks without relying on task-specific neural ISPs or full-model fine-tuning. By introducing a lightweight preprocessing module and efficient adaptation mechanisms (LoRA and prompt-based tuning), the proposed framework significantly reduces training and deployment costs.
- One of the key strengths lies in the ability to handle multiple vision tasks—detection, segmentation, and pose estimation—within a single, unified framework. This versatility is achieved without retraining the entire backbone, which is an important contribution toward general-purpose visual perception from RAW inputs.
- The experimental validation is thorough and well-structured. It spans several real and synthetic datasets with diverse sensor characteristics and lighting conditions. Results show strong performance and consistent improvements across tasks and exposure scenarios.
- The system is modular and well-motivated, making it practical for integration into real-world applications. The separation between sensor-specific preprocessing and task-specific adaptation is conceptually clean and promotes reusability.
- The paper follows good practices in architectural transparency and experimental design. The ablation studies are useful, and the authors are clear about the efficiency gains in terms of trainable parameters.

Weakness:
I have some minor concerns with regard to following points.
- The technical novelty of the paper is somewhat limited. While the integration of existing ideas (e.g., LoRA, prompt tuning, lightweight U-Nets) into the RAW domain is thoughtful and effective, the algorithmic innovation is not clearly highlighted.
- While the paper claims that the preprocessing module (sensor + illumination mapping and re-mosaicing) enables consistent representations across sensors, this claim is mostly supported by qualitative visualizations (e.g., PCA plots) rather than quantitative alignment metrics. A clearer evaluation of distribution alignment across sensor domains would strengthen the argument.
-  re-mosaicing block and its use of simple gating are introduced as key components, but there is no ablation or comparison against more conventional alternatives (e.g., using fixed Bayer demosaicing). This makes it difficult to assess whether these design choices are essential.
- Although the framework is task-agnostic in principle, each task still requires some adaptation via LoRA and SPP modules. The extent to which these components generalize across tasks or unseen sensor types is not deeply explored. In this sense the novelty in the paper sounds slightly overclaimed.

---

> ### Author Rebuttal · Authors · 2025-07-29
>
> We are very grateful for the insightful comments and valuable suggestions. The reviewer's feedback has provided us with a clear roadmap for strengthening the paper's clarity and impact.
>
> **Weakness 1**: We agree that the core novelty of Dr. RAW lies not in the invention of the individual PEFT techniques, but in their novel, synergistic integration to create a new, efficient, and high-performing paradigm for RAW-based vision tasks. As reviewer p2T5 astutely noted, our contribution is a complete, RAW-specific pipeline that thoughtfully combines these components to solve problems unique to this domain.
> Our specific novel contributions, which we will highlight more explicitly in the revised manuscript, are:
> 1. A New Paradigm for the RAW Domain: Dr. RAW challenges the standard approach in RAW-based vision, which typically requires full fine-tuning of both the network and complex, hand-crafted, or learnable ISP pipelines. Our work is the first to systematically apply and tailor modern PEFT techniques to address the challenges inherent in RAW processing, such as sensor and illumination variance. We demonstrate that a frozen, general-purpose backbone can achieve SOTA results on diverse RAW datasets by using a combination of lightweight pre-processing and parameter-efficient adaptation, a significant methodological contribution.
> 2. Domain-Specific Prompt Engineering (SPP): Our Sensor Prior Prompts (SPP) are a novel, domain-specific application of prompt tuning. Unlike generic visual prompts, SPP is uniquely designed to translate physical camera properties (the sensor and illumination matrices) into learnable embeddings that condition the model. This targeted injection of sensor-aware conditioning into a frozen backbone is a key innovation for handling data inconsistency in RAW files.
> 3. Synergistic integration for SOTA Performance: We demonstrate that the synergy between our lightweight RAW pre-processing, the global conditioning from SPP, and the targeted, task-specific updates from PEFT leads to SOTA performance with remarkable efficiency. The SPP steers the model based on sensor characteristics, while LoRA fine-tunes key operations for the specific downstream task. This combined strategy allows us to outperform fully-tuned models while updating only a very small fraction of the parameters.
>
> It is this specific, multi-stage integration that allows Dr. RAW to outperform fully-tuned methods while updating as little as 2% of the backbone's parameters.
> Following this valuable feedback, we will revise the Introduction and Related Work sections to more explicitly frame our contribution in this context, ensuring the novelty of our integrated framework is clear.
>
> **Weakness 2**: We thank the reviewer for their valuable feedback and the suggestion to strengthen the quantitative evidence for our pre-processing module's effectiveness.
> We would like to respectfully clarify that, in addition to the qualitative PCA visualizations, we include a quantitative analysis of the distribution alignment in Section 4.7, "Impact of Pre-processing Block." We recognize this section may not have been prominent enough, and we will ensure it is highlighted more clearly in the revised manuscript.
>
> To directly address the reviewer's point, our claim of achieving more consistent representations is supported by the following quantitative metric presented in the paper: We quantitatively measure the compactness of the chromaticity distribution before and after our pre-processing module, measured via reduced intra-class distance. The average intra-class distance decreases from 0.380 for the original RAW images to 0.259 for the mapped images. This 32% reduction is a direct, quantitative measure of improved distribution alignment, demonstrating that our module brings representations from varied sensor and lighting conditions into a tighter, more consistent cluster.
>
> This quantitative reduction in sample variability is not merely a statistical observation. As shown in our ablation study (Table 6), introducing the pre-processing block alone yields substantial gains, boosting object detection performance on LOD from 43.6 to 57.9 mAP. This demonstrates that the quantitatively measured improvement in distribution alignment is crucial for enabling more stable and effective learning.
> In the final version of the paper, we will create a dedicated table to present these quantitative metrics alongside the PCA plot to make the evidence for our claim more explicit and unmistakable. Thank you for helping us improve the clarity of our paper.
>
> **Weakness 3**: We thank the reviewer for this excellent point. While not explicitly labeled as an ablation, the comparisons against the "Default ISP" and "Demosaicing" baselines in our experimental tables (e.g., Table 2) are intended to serve this exact purpose.
> 1. Conventional Alternatives: The "Default ISP" and "Demosaicing" methods represent the conventional approach. They use a standard, fixed demosaicing algorithm as the primary pre-processing step before feeding the data to a fully-tuned network.
> 2. Our Learnable Approach: Dr. RAW replaces this with a more sophisticated, learnable pre-processing stage, where the re-mosaicing block is a key component for refining the image and correcting artifacts.
>
> The performance gap across all conditions validates the effectiveness of our design. For example, on PASCAL-RAW dataset, our method achieves 90.4 mAP (normal) and 89.7 mAP (dark), substantially outperforming “Default ISP” (86.7 mAP on normal light) and "Demosaicing" baseline (88.1 mAP on normal light, 86.5 mAP on dark light). This demonstrates that our learnable re-mosaicing block is an essential component.
>
>
> **Weakness 4**: Thank you for this comment, which allows us to clarify the nature of our contribution. We agree that task-specific adaptation is required. In fact, enabling this adaptation in a very efficient manner is our core novelty. The "task-agnostic" aspect of our framework refers to the frozen backbone and the pre-processing architecture. Unlike prior methods that require a fully fine-tuned, task-specific network for each application, our approach utilizes a single, unified backbone for all tasks. The lightweight modules, LoRA and SPP, are not a limitation of this approach but rather the highly efficient mechanism that makes it possible, steering the powerful, general backbone for a specific task.
>
> To directly evaluate the generalization to unseen sensors, we conducted a new zero-shot experiment. We took our model trained on the PASCAL RAW dataset (captured with a Nikon DSLR camera) and tested it without any fine-tuning, on RAW images of the same scenes captured with two unseen sensors, iPhone X and Samsung (we adopt methods [1] to do RAW-to-RAW mapping). The model achieves 88.3 mAP and 88.8 mAP, respectively. Compare with original performance 90.4 on Nikon, the results show only a marginal performance drop when tested on unseen sensors, demonstrating Dr.RAW's generalization.
>
> **Question 1**: Please refer to our response to **Weakness 3**.
>
> **Question 2**: We kindly refer the reviewer to our response to the above **Weakness 2**.
>
> **Question 3**: We thank the reviewer for this excellent question. The reviewer is correct that the pre-processing block's function is dependent on domain-specific characteristics like sensor response and illumination (though the zero-shot performance on unseen sensors is still outstanding, as shown in our response to **Weakness 4**). For this reason, the pre-processing block is not a fixed, universally shared component. To directly answer your questions: The pre-processing parameters are reusable for a specific trained task, but are fine-tuned for each new task. Domain-specific fine-tuning of this block is always part of our adaptation strategy for a new task. We will revise the manuscript to clarify which components are frozen versus which are fine-tuned per task.
>
> **Question 4**: Thank you for your feedback. This is an excellent question regarding why we chose this principled formulation over purely learned prompts.
> Our design is a deliberate trade-off aimed at maximizing generalization and efficiency while retaining a degree of interpretability. By grounding the prompts in the physical properties of the camera, we provide the model with a strong, structured prior. This helps the model generalize more effectively to new sensors and lighting conditions, as it doesn't need to learn these complex physical relationships from scratch. A purely random, learnable prompt would lack this physical grounding.
>
> To quantitatively validate this design, we conducted a new ablation study comparing our SPP against a baseline using an equivalent number of purely random, learnable embeddings (as in standard VPT). As shown in the table below, our SPP consistently outperforms the purely learned prompts, confirming the benefit of our physically-grounded approach.
> | Method                   | Dataset             | mAP  |
> |--------------------------|----------------------|------|
> | Learned Prompts (Baseline) | LOD                | 70.6 |
> | SPP (Ours)               | LOD                  | 72.1 |
> | Learned Prompts (Baseline) | PASCAL RAW (dark) | 88.3 |
> | SPP (Ours)               | PASCAL RAW (dark)    | 89.7 |
>
> We will add this discussion and the quantitative results to the final manuscript to make our design choices and their empirical validation perfectly clear. Thank you for helping us strengthen the paper.
>
> **Question 5**: We respectfully refer the reviewer to our response to **Weakness 4** above.
>
> [1]. Generalizing ISP Model by Unsupervised Raw-to-raw Mapping ACM MM 2024

---

> > ### Comment · Reviewer_uAEX · 2025-08-07
> >
> > Thank you very much for your detailed and thoughtful responses inculuding additonal evaluations. Most of my concerns have been clarified, and I believe the responses have provided sufficient information for further discussions between the AC and reviewers.
> >
> > I would like to ask one follow-up question. The reported zero-shot results on iPhone and Samsung sensors are encouraging. However were the illuminants in the unseen test sets similar to the ones seen during training or completely different? I simply want to learn how the zero-shot capability is affected by the lighting condtion.

---

> > > ### Author Response · Authors · 2025-08-07
> > >
> > > Dear reviewer, we greatly appreciate your recognition of our rebuttal and your helpful suggestions.
> > >
> > > Regarding the results on iPhone and Samsung sensors, the illuminants in the test sets are completely different from those in the training set. The training was still conducted using RAW images from Nikon-RAW, while the testing was performed on converted iPhone-RAW and Samsung-RAW. You may consider this as a fully out-of-domain (OOD) zero-shot performance across illumination scenarios.

---

> > > > ### Comment · Reviewer_uAEX · 2025-08-07
> > > >
> > > > Thank you for the clarification. It's impressive to see that the evaluation was conducted under such challenging out-of-domain conditions.

---

### Official Review · Reviewer_p2T5 · 2025-07-04

**Clarity:** 3
**Significance:** 3
**Originality:** 3
**Rating:** 4
**Confidence:** 4

**Summary:**

This paper introduces Dr. RAW, a unified and parameter-efficient framework for performing high-level computer vision tasks directly on RAW image data. The work addresses two key challenges in RAW-based vision: data inconsistency arising from different camera sensors and lighting conditions, and tuning inefficiency caused by the need to fully fine-tune entire models (both ISP and backbone) for each specific task.

**Questions:**

See weaknesses

**Ethical Concerns:**

["NO or VERY MINOR ethics concerns only"]

**Limitations:**

See weaknesses

**Paper Formatting Concerns:**

See weaknesses

**Quality:**

3

**Strengths And Weaknesses:**

Strengths
- Comprehensive Evaluation: The experimental validation is extensive and a major strength. The authors evaluate their method on four different and representative high-level tasks, using a total of nine datasets that cover various lighting conditions (normal, dark, over-exposed). This thoroughness strongly supports the claims of generality and robustness.
- State-of-the-Art Performance with High Efficiency: The core contribution—achieving superior or competitive performance while tuning only a tiny fraction of parameters—is highly compelling. The results, effectively summarized in the radar chart (Fig. 1) and detailed in Tables 1-5, show that Dr. RAW consistently outperforms prior methods.
Weaknesses
- Clarity on the Sensor Prior Prompt (SPP) Mechanism: The SPP module is a central piece of the proposed adaptation strategy, yet its mechanism could be explained more intuitively. The paper states that it projects the concatenation of the sensor and illumination matrices into embeddings (Eq. 3), but the intuition behind how this effectively injects "sensor-aware conditioning" could be elaborated. A small diagram illustrating how these low-dimensional prompts interact with image patch embeddings within the attention mechanism would be highly beneficial for the reader's understanding.
- Dependency on Large-Scale RAW Pre-training: The best results are achieved with a Swin Transformer pre-trained on the ADE20K RAW dataset. This is a crucial detail and a potential practical limitation. The paper would be stronger if it discussed the implications of this dependency. For example: How does the method perform if such a large-scale RAW dataset is not available? How much of the performance gain is attributable to the RAW pre-training versus the proposed adaptation modules? While Table 7 compares to an ImageNet-1k pre-trained backbone, a more explicit discussion of this limitation and potential future work to mitigate it would be valuable.
- More Granular Ablation of the Pre-processing Block: The pre-processing stage consists of three components: a sensor mapping block, an illumination mapping block, and a U-Net-like re-mosaicing block. The current ablation study (Table 6) treats this entire stage as a single unit. A more fine-grained ablation study that dissects the individual contribution of each of these three sub-components would provide deeper insights into which parts are most critical for handling data inconsistency. This would further strengthen the design choices presented.
- Positioning of Novelty: The core adaptation techniques, LoRA and prompt tuning, are borrowed from the existing PEFT literature. The paper does a good job of citing these works. However, it could be slightly more explicit in the introduction or related work that its primary novelty lies not in inventing these PEFT techniques, but in the novel synergistic application and integration of these methods within a RAW-specific pipeline. This would help to precisely position the paper's contribution in the context of both RAW processing and efficient model adaptation.

---

> ### Author Rebuttal · Authors · 2025-07-29
>
> We sincerely thank the reviewer for the time and effort in providing a thorough and constructive review of our manuscript. The feedback has been invaluable in helping us identify areas for improvement, and we have addressed each point in detail below.
>
> **Weakness 1**: We appreciate the reviewer’s feedback on the clarity of the Sensor Prior Prompt (SPP) mechanism. We agree that a more intuitive explanation of how SPP injects "sensor-aware conditioning" benefits the paper, and we will include a more detailed figure in the final manuscript.
> The idea behind SPP is partly inspired by VPT [1], and we are the first to apply this concept to RAW-based vision tasks. The goal of SPP is to adapt a large, pre-trained backbone to new tasks or data distributions with minimal changes to the model itself. Instead of fine-tuning all the weights, which is computationally expensive and requires storing a separate model for each task, SPP introduces a small set of learnable parameters called "prompts" that are prepended to the input data. The backbone remains frozen, and only these prompts and a task-specific head are trained.
>
> We use the sensor and illumination matrices ($M$ and $L$) as a compact "fingerprint" of the capture conditions. This low-level fingerprint is translated into a high-level "hint" to guide the backbone, enabling it to adapt its processing in a sensor & lightness-aware manner.
>
> The process involves two key steps: **Projection** and **Interaction**.
> 1. Learnable Projection: As noted in Eq.3, we concatenate the embeddings projected from $M$ and $L$ using an FFN. The FFN acts as a learnable, non-linear mapping. It learns the optimal way to translate the matrix values into the high-dimensional embedding space that the transformer operates in. The FFN's weights are updated during training, and the model learns to create prompts that are maximally useful for the downstream task. It isn't a fixed projection; it's an optimized one that learns to emphasize the most important sensor characteristics for perception.
>
> 2. Interaction within Self-Attention: Once generated, the prompts are injected into the backbone by prepending them to the sequence of image patch embeddings. (Fig. 7 in Appendix B.3 illustrates the interactions within self-attention). This is where the "conditioning" happens. The self-attention computes relationships between all tokens in its input sequence. By adding our prompts, we enable three critical interactions, as described in and derived from Eq. 5:
>     1. Prompts attend to Image Patches: The prompts can gather context from the entire image, allowing the model to understand how the global sensor properties relate to specific image content.
>     2. Image Patches attend to Prompts: Each image patch can "query" the sensor-specific information held in the prompts. This allows local features to be interpreted differently based on the global sensor fingerprint. For example, the model might learn to process a noisy blue channel in a patch differently if the prompts indicate it came from a sensor known to have such characteristics.
> Prompts attend to Prompts: The prompts interact with each other to form a coherent, internal representation of the sensor's properties before they influence the image patches.
>
> **Weakness 2**: Thanks for the advice, we agree that the dependency on large-scale RAW pre-training is an important practical consideration. Our Dr. RAW is robust enough to perform well even with smaller RAW pre-training datasets, and our adaptation modules provide significant gains on their own. We acknowledge that the availability of large-scale RAW datasets is a practical consideration, and we've conducted a new ablation study to directly address your questions. We experimented with the scenario where large-scale synthetic RAW data like ADE20K-RAW is not available. We pre-trained a new backbone using only the much smaller PASCAL-RAW dataset (4259 images). We then evaluated this backbone on the LOD and LIS tasks. The results are as follows:
>
> | Downstream Task Dataset | Backbone Pre-training Dataset             | mAP  |
> |-------------------------|-------------------------------------------|------|
> | LOD                     | ADE20K (RAW, 20210 images)                | 72.1 |
> | LOD                     | PASCAL-RAW (RAW, 4259 images)             | 69.3 |
> | LOD                     | in1K (RGB, 1281167 images)                | 67.8 |
> | LIS                     | ADE20K (RAW, 20210 images)                | 41.2 |
> | LIS                     | PASCAL-RAW (RAW, 4259 images)             | 37.8 |
> | LIS                     | in1K (RGB, 1281167 images)                | 35.9 |
>
> This new study shows that while large-scale pre-training yields the best results, our method still achieves strong performance when pre-trained on a smaller, more accessible RAW dataset.
> Attributing Performance Gains: The performance gains come from both the RAW pre-training and our adaptation modules.
> 1. Gain from RAW Pre-training: The new results show that pre-training on even a small RAW dataset (PASCAL-RAW) boosts performance to 69.3 mAP, better than using a standard ImageNet-1k (RGB) pre-trained backbone (67.8 mAP) shown in Table 7. Using a large-scale RAW dataset (ADE20K RAW) provides an additional lift to 72.1 mAP.
> 2. Gain from Adaptation Modules: As shown in Table 6, our adaptation modules alone provide a substantial lift. Our method achieves 72.1 mAP on LOD, which is a massive improvement over a simple frozen baseline (43.6 mAP). This shows our modules are highly effective at bridging the domain gap.
> 3. Future Work: We sincerely appreciate the advice and agree that this is a limitation and will add a discussion to the paper. A promising direction for future work is to explore self-supervised pre-training on large, unlabeled RAW image collections. This could further enhance performance and reduce the dependency on large-scale labeled RAW datasets.
>
> **Weakness 3**:
> Thank you for pointing this out. We agree that a more granular ablation study provides more insights. We conducted new experiments, and the results are shown in the table:
>
> | Re-mosaicing | Sensor Mapping Block | Illumination Mapping Block | LOD  | PASCAL RAW (normal) | PASCAL RAW (over-exp) | PASCAL RAW (dark) |
> |--------------|----------------------|----------------------------|------|----------------------|------------------------|--------------------|
> |              |                      |                            | 43.6 | 81.4                 | 86.0                   | 59.9               |
> |              |                     |        √                    | 50.8 (+7.2) | 84.1  (+2.7)               | 87.5  (+1.5)                 | 66.6  (+6.7)             |
> |              |           √           |                            | 48.8 (+5.2) | 83.8  (+2.4)               | 87.7  (+1.7)                 | 68.3   (+8.4)            |
> |              | √                    |     √                       | 55.4 (+11.8) | 86.2  (+4.8)               | 88.1  (+2.1)                 | 74.7  (+14.8)             |
> |      √        |                     |                           | 53.2 (+9.6) | 84.9 (+3.5)                | 87.1  (+1.1)                 | 71.9   (+12.0)            |
> | √            | √                    | √                          | 57.9 (+14.3) | 88.6  (+7.2)               | 89.5  (+3.5)                 | 81.1  (+21.2)             |
>
> This detailed analysis offers several key insights: Both the Sensor Mapping and Illumination Mapping blocks individually provide a significant performance improvement over the baseline. And their combined effect is synergistic, yielding a larger gain than either block alone, which validates the design of our mapping stage. The re-mosaicing block also offers an individual contribution, confirming its importance. The best performance is achieved when all three components are used together, demonstrating that each part of our pre-processing pipeline is essential for the final SOTA results.
>
> **Weakness 4**: Thank you for this constructive feedback. We agree that a more explicit framing of our contribution will strengthen the paper.
> We will clarify our specific contributions in the revised manuscript as follows:
> 1. A novel application of PEFT to the RAW Domain: To our knowledge, this is the first work to systematically apply and tailor modern PEFT techniques to address the challenges inherent in RAW processing, such as sensor and illumination variance. While prior RAW-based methods often rely on fully tuning dedicated ISP modules and the entire network backbone, we demonstrate a new, more efficient path. We show that a frozen, pre-trained backbone can be successfully adapted, which represents a significant methodological shift in this domain.
> 2. The design of SPP: Our SPP module is a novel, domain-specific adaptation of VPT. Unlike general-purpose prompts, which are typically independent learnable parameters, SPP is uniquely designed to translate the physical properties of the camera into conditional "hints" for the model. This targeted injection of sensor-aware conditioning into a frozen backbone is a key novel component of our framework.
> 3. Synergistic integration for SOTA Performance: We demonstrate that the synergy between our lightweight RAW pre-processing, the global conditioning from SPP, and the targeted, task-specific updates from LoRA leads to SOTA performance with remarkable efficiency. The SPP steers the model based on sensor characteristics, while LoRA fine-tunes key operations for the specific downstream task. This combined strategy allows us to outperform fully-tuned models while updating only a fraction of the parameters.
>
>
>
>
> [1]. Jia, Menglin, et al. "Visual prompt tuning." European conference on computer vision. Cham: Springer Nature Switzerland, 2022.

---

> > ### Comment · Area_Chair_E9BG · 2025-08-08
> >
> > AC would like to thank authors' for their rebuttal. Dear Reviewer p2T5, have your previous concerns regarding the clarity and the novelty of its contributions been adequately addressed? Your invaluable feedback on this paper is greatly appreciated.

---

### Note · Authors · 2025-08-15

We sincerely thank the reviewers and the Area Chair for their time and for providing a highly constructive review process. The feedback was invaluable in helping us clarify our core contribution: a new, more effective paradigm for RAW-based high-level vision tasks (including object detection, semantic segmentation, instance segmentation and pose estimation). Our framework demonstrates the benefits of combining in-domain RAW-based pre-training with sensor prior knowledge. It outperforms both the traditional ISP-based approach and other fully fine-tuning RAW-based methods, while simultaneously addressing $data$ $inconsistency$ and $tuning$ $inefficiency$.

To substantiate our claims and address the points raised, we conducted several new experiments during the rebuttal period. We have now provided:

1. Zero-shot generalization tests showing that our model maintains high performance on unseen sensors (out-of-domain evaluation).

2. Detailed ablation studies validating each component of our pre-processing blocks, including the Re-mosaicing Block, Sensor Mapping Block, and Illumination Mapping Block.

3. Experiments demonstrating that our method remains highly effective even when pre-trained on an sRGB dataset (i.e., ImageNet-1K) or on a smaller, more accessible RAW dataset (i.e., PASCAL RAW).

We appreciate the reviewers’ professional suggestions, which have made us more confident that the manuscript is now stronger and more comprehensive as a result of this discussion. These new results, alongside our detailed clarifications, directly address the concerns regarding generalization, the fairness of our experimental comparisons, and the novelty of our integrated approach. We are committed to incorporating all of these clarifications, analyses, and new results into the final version.

---

### Decision · Program_Chairs · 2025-09-17

**Decision:**

Accept (poster)

**Comment:**

This paper is reviewed by four reviewers. All the 4 reviewers recommend acceptance (3 borderline accepts and 1 accept). The authors have comprehensively addressed the weaknesses and questions raised by the reviewers, strengthening the paper toward acceptance. These include clarifications regarding the novelty, explanations of the learnable re-mosaicking block and LoRA, and the conducted zero-shot and ablation studies for the proposed pre-processing blocks. Given the thorough rebuttals and demonstrated improvements, the paper is well-positioned for acceptance.

The authors should include the additional experiments (e.g., zero-shot results) and add clarifications to address the raised concerns on effectiveness, novelty and clarity in the final copy as promised. The recommendation is acceptance.